# Modelling the impact of behavioural interventions during pandemics: A systematic review

**Tsega Kahsay Gebretekle** [ID]*, **Casper Albers** [ID]

Department of Psychometrics & Statistics, University of Groningen, Groningen, The Netherlands

* tsegaka@yahoo.com

## Abstract

### Background

Many studies examined the impact of behavioural interventions on COVID-19 outcomes. We conducted a systematic review to gain insight into transmission models, following PRISMA 2020 guidelines. We included peer-reviewed studies published in English until December 31, 2022, focusing on human subjects, modelling, and examining behavioural interventions during COVID-19 using real data across diverse geographical regions.

### Methods

We searched seven databases. We used descriptive analysis, network analysis for textual synthesis, and regression analysis to identify the relationship between the basic reproduction number $R_0$ and various characteristics. From 30, 114 articles gathered, 15, 781 met the inclusion criteria. After deduplication, 7, 616 articles remained. The titles and abstracts screening reduced these to 1, 764 articles. Full-text screening reduced this to 270, and risk-of-bias assessment narrowed it to 245 articles. We employed combined criteria for risk of bias assessment, incorporating domains from ROBINS-I and principles for modeling.

### Results

Primary outcomes focused on $R_0$, COVID-19 cases, and transmission rates. The average $R_0$ was 3.184. The vast majority of studies (90.3%) used compartmental models, particularly SEIR models. Social distancing, mask-wearing, and lockdowns were frequently analyzed interventions. Early and strict implementation of these interventions significantly reduced transmission rates. Risk of bias assessment revealed that 62.6% of studies were of low risk, 24.1% moderate, and 9.3% high risks. Common issues included transparency, attrition bias, and confounding factors.

### Conclusions

This comprehensive review highlights the importance of behavioural interventions in reducing COVID-19 transmission and areas for improving future research transparency and robustness. Our risk of bias criteria offers an important framework for future systematic

**Data Availability Statement:** Any data used for this study such as the extracted data are available on the Open Science Framework (OSF) at the following view-only link: https://osf.io/dy9ck/?

view_only=
b84db0290b8e46839d2704d6136b29a0. They are
all in Excel file format.

**Funding:** C.J. received the grand. This study is
funded by ZonMW ('BEhavioural and social
sciences and pandemic PREPAREDness', grant
number 10710022210002). The url for the funder
is: https://www.zonmw.nl/nl. The sponsors or
funders did not play any role in the study design,
data collection and analysis, decision to publish, or
preparation of the manuscript.

**Competing interests:** The authors did not have any
competing interests.

reviews in modeling studies of interventions. We recommend that future studies enhance
transparency in reporting and address common biases such as attrition and confounding.

## Introduction

### Rationale

Pandemics have a significant global impact, including health, social, political, and economic
impacts [1]. The COVID-19 pandemic has caused a severe global recession, resulting in wide-
spread job losses and economic inequality. It is crucial to understand the role that various
behavioural interventions can play in shaping the trajectory of disease outbreaks and building
resilience, a critical step in preparing for future pandemics. Besides the economic effects and
the effects on people's physical health, the pandemic also harmed citizen's mental health. Plans
and actions are needed to strengthen healthcare systems globally and to restore economies and
societies damaged by COVID-19. This requires collaborative work [2–5].

One of the early challenges during COVID-19 was monitoring social and behavioural indi-
cators to assess the effectiveness of prevention measures. Social and behavioural interventions
refer to strategies aimed at influencing individual and group behaviours to reduce disease
spread and promote health. Examples of such measures are including mask-wearing, social
distancing, hand hygiene, and contact tracing. However, limited methodological evidence cur-
rently exists on integrating behavioural data into epidemiological models for pandemic spread
and assessing the impact of behavioural interventions on transmission and hospitalization
rates [6–8]. This gap underscores the need for specialized modelling approaches.

To address the challenges, a systematic literature search was conducted. This systematic
review was conducted to generate a comprehensive overview of transmission models including
specific transmission behaviours related to the COVID-19 pandemic and how the associated
model parameters are informed by behavioural data. Understanding these factors is crucial for
developing evidence-based strategies to mitigate the impact of future pandemics. As a result,
the task helps (future) social and behavioural researchers to assist and advise policymakers in
the future more effectively.

### Objectives

The primary objective of this review is to explicitly address key questions related to the impact
of behavioural interventions during COVID-19. Further, it will advance science by providing a
foundation for selecting and advancing models that can be used effectively in exploratory sim-
ulation studies. These studies will play a critical role in estimating the impact of behavioural
interventions on pandemic outcomes. Through this comprehensive approach, the systematic
review aims to provide valuable insights to support subsequent modeling efforts and advance
our understanding of the relationship between behavioural interventions and COVID-19 out-
comes. To generate important models that embedded behavioural aspects, various models
were evaluated based on predefined criteria. These criteria include the ability of the models to
link behaviours to transmission and outcomes [9], identify subgroups, including individual
heterogeneity and dynamics in behaviour [10], incorporation of regional or temporal differ-
ences [11], availability of open source software [12], and model performance/validation [13].

### Research question

The main research question of this paper is "Under what conditions can behavioural interventions impact COVID-19 outcomes?". According to the PICO framework [14], the review focused on participants, interventions, comparators, and outcomes. These terms are described as follows:

- **Participants**: Individuals exposed to behavioral interventions to prevent or mitigate the spread of COVID-19, such as hand washing, mask-wearing, social distancing, and contact tracing. Studies that have participants from all types of backgrounds; regardless of age, gender, race, ethnicity, socioeconomic status, or health status will be included.

- **Interventions**: Behavioural interventions aimed at addressing or mitigating the impact of pandemics focusing on a) Enhancing testing and Isolation and b) Vaccination.

- **Comparators**: Situations where these behavioural interventions are not implemented or are less effective. Example: Social distancing (implemented consistently vs. inconsistently or not at all).

- **Outcome**: COVID-19 cases, deaths, transmission rates, basic reproduction number ($R_0$), and impact of interventions on COVID-19 outcomes, considering factors like effect size and the degree to which the intervention penetrates particular population subgroups.

By providing explicit statements for each component of the PICO framework, this review aims to systematically address the complex interplay between behavioural interventions and COVID-19 outcomes, contributing to the advancement of knowledge in this critical area.

## Methods

The Preferred Reporting Items for Systematic Reviews and Meta-Analyses statement (provided in S2 Appendix) was used to review different research papers [15].

### Eligibility criteria

The study population of interest encompasses individuals who have been exposed to behavioral interventions aimed at preventing the spread of COVID-19. The interventions under investigation include changes in hygiene practices, social distancing, wearing masks, isolation behaviours, and vaccination adherence.

Peer-reviewed articles published between January 1, 2019, and December 31, 2022, were considered to capture the evolving landscape of behavioural interventions during the COVID-19 pandemic. Only articles written in English were included. The review emphasizes articles published in various countries, irrespective of region, with a specific focus on behavioural interventions. Articles that include humans and the behaviour of humans regarding COVID-19 were included, which indicates that researches on plants and animals were excluded. The contents of the included articles comprise a variety of research papers with real data, such as original research articles, published reports, and conference papers/proceedings. Articles must also include transmission models relevant to COVID-19 and incorporate behavioural interventions as a key aspect. Additionally, the inclusion criteria for articles in this systematic review were assessed based on their relevance to the research question and the validity of their methodologies.

Exclusion criteria involve non-English articles, book chapters, letters, editorials, comments, retracted papers, short surveys, notes, review articles, systematic reviews, and qualitative

studies. These criteria aim to ensure a comprehensive and focused approach to understanding the impact of behavioural interventions on pandemic outcomes.

## Information sources

To ensure a comprehensive approach to data collection, we used seven databases, each serving a specific purpose:

1. PsychInfo and Psychology and Behavioural Sciences Collection: We included these databases because both are valuable sources of behavioural data. PsychInfo is the largest resource devoted to peer-reviewed literature in behavioural science and mental health. Similarly, the Psychology and Behavioural Sciences Collection covers information concerning topics in emotional and behavioural characteristics, psychiatry & psychology, mental processes, anthropology, and observational and experimental methods.

2. MathSciNet via EBSCOhost: To cover the mathematical literature, we incorporated MathSciNet, offering access to a carefully maintained and easily searchable database of reviews, abstracts, and bibliographic information.

3. Web of Science and Scopus: We included both databases to ensure a comprehensive and diverse coverage of research literature across various disciplines. Web of Science provides a complementary perspective on research across various domains. Similarly, Scopus is a valuable resource for multidisciplinary research.

4. MEDLINE and EMBASE: The inclusion of these databases ensures comprehensive coverage of biomedical and pharmaceutical literature. MEDLINE provides extensive coverage across various biomedical topics, including medicine, nursing, dentistry, veterinary medicine, and healthcare systems. EMBASE is known for its focus on drug development and clinical pharmacology.

Additionally, Google Scholar was utilized for a manual search of papers.

The search for studies began on September 15, 2023, and the initial search period ended on October 6, 2023. After consultation with consortium members on November 24, 2023, modifications were made to the keywords used. Subsequently, a second search started on November 24, 2023, and the last search was carried out on November 28, 2023.

## Search strategy

The search strategy focused on the following keywords:

'("COVID*" OR "corona*")' AND '("Model*")' AND '("Transmission Model*" OR "Compartmental Model*" OR "Population Dynamic*" OR "Epidemiological Model*" OR "Mathematical Model*")'

This search query was used across the four databases: PsycInfo, MEDLINE, Psychology and Behavioural Sciences Collection, and MathSciNet.

For EMBASE, Web of Science, and Scopus some technical alterations to this search query were needed. These alterations are explained in S1 Appendix.

## Selection process

We performed a four-step screening process to identify relevant articles for inclusion in the systematic review. The first step involved gathering all articles that matched the search terms.

In the second step, we identified and removed duplicate articles. In the third step, we examined the titles and abstracts of articles to select those focused on behavioral interventions and real data. In the fourth step, the contents of the full articles were assessed as a comprehensive full-text review to decide whether they should be included or excluded based on their relevance to the study. These studies form the basis for data extraction and subsequent analysis in the systematic review.

The selection process involved both authors who reviewed independently and assessed each study's eligibility based on the predetermined criteria outlined in the Eligibility criteria subsection. Initially, TKG screened titles and abstracts to identify potentially relevant studies. Subsequently, CJA independently cross-checked a random selection of the identified studies by reviewing the titles, abstracts, and eligibility criteria. Subsequently, full-text articles were retrieved and independently assessed for eligibility. Any discrepancies or uncertainties were resolved through discussion and consensus between the two reviewers. Rayyan.ai assisted us in the screening process, facilitating collaboration between reviewers, streamlining the tagging process, and providing the option to prioritize articles based on keywords for inclusion [16]. Rayyan.ai was also used to store data, including detailed citation information, abstracts, and key outcomes identified during the screening process. The screening process utilized classification by Rayyan.ai into 'include', 'exclude', and 'maybe'. All papers in the 'maybe' group were included for the full-text review phase [16]. EndNote was used to store and do the duplicate detection process as well as reference management software [17]. All articles found from the database searches were imported into EndNote for the de-duplication process. The articles after the duplicate detection process were imported into Rayyan.ai for the title and abstract screening process. Excel sheets were used to perform the data extraction process outlined in the Data items subsection.

## Data collection process

Data extraction followed standardized and piloted data extraction forms to ensure comprehensive coverage of relevant information. The form was designed to capture relevant information, including study characteristics explained in the Data items subsection.

As a pilot extraction process, five articles were extracted, and both authors discussed the extracted elements to ensure consistency and mutual understanding. Any disagreements were resolved through discussion. Following the pilot phase, the full extraction process commenced. As part of the extraction process, relevant data were collected directly from the reports by reviewing key sections such as methods, results, and conclusions.

## Data items

A standardized form was used for data extraction of characteristics of studies, outcomes, and risk of bias. We extracted the data using the following sample characteristics or items: authors, digital object identifier (DOI), title, year of publication, country of data collection, model name, study design, sample size, type of data (primary data, secondary data, experimental data, or other), target population, setting, intervention type, outcome measure, basic reproduction Number ($R_0$), effective reproduction number($R_{eff}$ or $R_c$), outcome measure results, key findings, additional comments, exact population size consideration ($N = 0$ as a small setting, $N = 1$ as the whole population), whether a compartmental model is used and whether the paper was published open access. The format of the data extraction columns used with extracted items is in Table 2 of Study characteristics. The full extracted data is provided in Excel format and can be accessed from the OSF link https://osf.io/dy9ck/.

The details of extraction items, their descriptions, and possible values are displayed in S1 Table. Each item corresponds to a specific aspect of the studies under review, providing a comprehensive framework for data extraction.

## Study risk of bias assessment

The authors assessed each study for risk of bias. TKG independently evaluated the risk of bias for all included studies and CJA took an independent sample of articles and independently evaluated the risk of bias. Both reviewers worked independently to minimize bias during the assessments. Any discrepancies between reviewers in risk of bias assessments or judgments were resolved through discussions.

We used Python and the "matplotlib" library to create risk-of-bias plots [18], where the look and style were adopted from the "robvis (visualization)" online tool [19].

In this systematic review, we assessed the risk of bias in the included studies based on combined approaches, considering both general and specific criteria to ensure a comprehensive evaluation of potential biases. From the guidance provided by the report "Guidance for the Conduct and Reporting of Modeling and Simulation Studies in the Context of Health Technology Assessment [20], we assessed the risk of bias considering eight principles. Additionally, we adopted seven domains such as selection bias, performance bias, detection bias, etc from the ROBINS-I (Risk Of Bias In Non-randomized Studies of Interventions) tool to align with established methods for bias assessment in systematic reviews [21]. We systematically applied these criteria to each study included in our review. For each domain and principle, we assigned a score of "Low Risk", "Moderate Risk", "High Risk" or "Unclear" risk of bias. A description of the eight principles and the seven adopted domains is presented/outlined in S2 Table.

**Rating the overall risk of bias level.** According to Cochrane guidelines [22], the overall risk of bias in a study is determined by assessing the relative importance of different domains, and not all domains carry equal weight in bias assessment. In our case, key domains, such as "Research Question, Goals, and Scope" (D1), "Data-Informed Model" (D4), "Attrition Bias" (D12), and "Confounding" (D14) are critical due to their direct influence on the study's foundational precision. Considering that two or more critical domains (such as D1, D4, D12, or D14) are rated high risk may suggest an overall high-risk judgment, as this would imply that major parts of the study's design are unreliable. Additionally, combined high risks in several domains tend to indicate that the study results are unreliable. An overall rating is classified as shown in S7 Table.

## Effect measures

In our systematic review, we focused on several outcomes related to the impact of behavioral interventions during pandemics. The key outcomes included death rates, COVID-19 cases, and reproduction number ($R_0$). Although we did not extract raw numeric values for death rates or transmission rates, we synthesized the effect measures narratively by reporting the qualitative and quantitative outcomes as stated in each study. For example, if a study reported a 20% reduction in death rate, this outcome is included in our extracted data. Each included study provided specific effect measures tailored to their respective outcomes and modeling approaches. Effect measures used in our synthesis included:

- $R_0$ **(Basic Reproduction Number)**: This measure was used to measure the average number of secondary infections produced by a single infected individual in a fully susceptible population. As a result of various behavioral interventions such as social distancing, quarantine, isolation, and contact tracing studies reported changes in $R_0$.

- **Death Rates**: Although raw death rates were not extracted, we synthesized findings based on reported changes in mortality attributed to behavioral interventions. Studies often reported relative reductions or trends in death rates following the implementation of interventions.

- **COVID-19 Cases**: Similar to death rates, we synthesized findings from studies that reported changes in the number of COVID-19 cases. This included reported percentage decreases, absolute reductions in case numbers, and other descriptive statistics provided.

## Synthesis methods

**Study selection process.**   In our study selection process, we established clear inclusion and exclusion criteria to ensure the relevance and quality of the studies included in our synthesis. We considered studies focused on modelling the impact of behavioral interventions during COVID-19 and reported outcomes such as death rates, number of COVID-19 cases, and reproduction number ($R_0$). For qualitative synthesis, narrative analysis, word clouds, and Network visualization were used. We used word clouds on the following items to identify recurring themes and patterns across studies: Model name, Outcome measure results, and Key Findings. For quantitative synthesis, descriptive statistics were used. We summarized the following study characteristics in tabular format: publication year, study location (continent), study design, sample size, type of data, basic reproduction number ($R_0$), population consideration($N = 0$ for a small setting, $N = 1$ for the whole population), compartmental? (Yes = 1, No = 0), and open access? (Yes = 1, No = 0). We did not perform a meta-analysis because of the heterogeneity in study designs, intervention types, outcome measures, and populations studied. The diverse nature of the data and the varying methodologies used across the included studies made it impossible to calculate a combined effect.

**Data preparation.**   We prepared our data to ensure the accuracy and reliability of our synthesis. We used the following Methods:

- **Handling missing data**: Missing values in numerical variables such as $R_0$ and sample size were addressed by examining the context in which these values were missing. When these variables were not explicitly provided in some studies, we did not simply discard these studies. Instead, we inferred them by considering other details and context in the study. This involved looking up the descriptions of the study population, the study's methodology, and the results sections. Additionally, we checked other external sources and supplementary materials such as GitHub codes that might indicate these variables to gather the required information.

- **Data standardization**: We systematically categorized and standardized textual variables for consistency across studies. This involved coding categorical variables such as study design ("modeling and simulation study" = 1, "observational study" = 2, and "Predictive modeling study" = 3), and type of data("experimental data" = 1, Mixed data = 2, "Primary data" = 3, "secondary data" = 4). For the variable *population consideration*, we coded 0 for studies that considered different subgroups as smaller settings whereas 1 for studies that considered the entire population, such as the population of a whole country. For instance, if a study included separate counts for nurses, doctors, and patients, we combined these subgroups and coded this variable as 0. For the variables *Compartmental* and *open access*, we used 0 to code 'No' and 1 to code 'Yes'. Similarly, in the risk of bias assessment data, we coded 1 for "low risk of bias", 2 for "moderate", 3 for "high", and 0 for "unclear".
The systematic categorization and standardization of textual variables facilitate comparative

analysis and thematic synthesis across diverse study contexts [23]. This process helps to identify common patterns, trends, and themes that may not be immediately apparent from the individual studies.

- **Sample size adequacy**: To ensure the generalizability of the study findings, we assessed the adequacy of the sample sizes reported in the included articles by comparing them with the population sizes of the respective study locations.

**Results tabulation and visualization.**   To effectively present and interpret the findings from our synthesis, We used descriptive statistics such as frequency tables and graphical representations to summarize the continuous and categorical variables. We employed VOSviewer [24] to extract terms from the title and abstract fields of the included articles, facilitating the creation of a network that represents the relationships between these terms. This method allowed us to identify the key concepts and themes within the body of literature under review, enhancing the synthesis and interpretation of study findings.

We also conducted a word cloud analysis using the "wordcloud" package in R [25] to visually represent the frequency of keywords from the included studies where the size of each word indicated its frequency of occurrence. This visualization provided a quick and intuitive overview of the predominant topics and themes within the literature, identifying key areas of focus and research trends.

**Results synthesis methods and rationale.**   We used a narrative approach to synthesize the results from the individual studies. That means, we carefully reviewed and summarized the key findings and methodologies used in each study. This allowed us to identify common themes, trends, and variations in the modeling approaches and their corresponding outcomes. The rationale for choosing this narrative synthesis method is that it provides a comprehensive and flexible way to integrate the findings from modeling studies with heterogeneous data sources, different model structures, and analytical techniques [26]. We ensured consistency in reporting and synthesis across studies by applying appropriate statistical methods.

**Exploration of heterogeneity.**   To explore possible causes of heterogeneity among study results, we conducted subgroup analyses based on key study characteristics, such as type of data (primary, secondary, experimental) and study design ("modeling", "modeling and simulation", "observational", and others). We also conducted a regression analysis considering the basic reproduction number($R_0$) variable as a dependent variable. We considered the following independent variables: population consideration, compartmental, type of data, sample size, study design, study location (continent), and open access.

**Sensitivity analysis description.**   We did not conduct formal sensitivity analyses on the overall narrative synthesis due to the qualitative nature of the approach.

## Reporting bias assessment

Due to the high heterogeneity observed and the lack of confidence intervals or variances for the reported $R_0$ values, traditional meta-analysis techniques such as funnel plots and Egger's test were not feasible. As an alternative, we conducted a qualitative assessment of reporting bias based on a comprehensive risk of bias evaluation. The risk of bias assessments were performed across 15 domains, where one of the domains is reporting bias. We evaluated the studies for the presence of reporting bias by analyzing the completeness and transparency of the reported outcomes. The findings of this assessment are detailed in the Reporting biases results subsection.

### Certainty assessment

The GRADE (Grading of Recommendations Assessment, Development, and Evaluation) approach was used to assess the overall certainty of evidence for each outcome of interest [27] Due to the use of narrative synthesis for most results, the certainty of the evidence may be downgraded due to limitations in combining data from different studies. This will involve considering five factors: overall risk of bias, imprecision, directness, heterogeneity, and publication bias, which is explained as follows:

- **Overall risk of bias**: This is the overall assessment of the risk of bias in the included studies.

- **Imprecision**: This is the uncertainty in the effect estimate due to the size and variability of the included studies.

- **Indirectness**: This is the extent to which the studies are similar to the target population and intervention of interest.

- **Heterogeneity**: This is the variability in the effect estimates between the included studies.

- **Publication bias**: This is the potential for studies with negative or non-significant findings to be less likely to be published.

    The certainty of evidence will be graded as:

- **High**: The evidence is strong and consistent.

- **Moderate**: The evidence is moderate or inconsistent.

- **Low**: The evidence is weak or conflicting.

- **Very low**: The evidence is very weak or very conflicting.

## Results

### Study selection

Fig 1 provides a flowchart of the article search and study screening. Initially, We searched articles across seven (7) databases and retrieved a total of 30, 114 articles. The inclusion and exclusion criteria, which involve filtering articles based on the criteria mentioned in Eligibility criteria Subsection, were applied during the first screening process. For example, retracted articles were identified and excluded due to their retraction status based on each database's allowance to do so. After the initial screening process, 15, 781 articles remained. However, after importing the 15, 781 articles into EndNote (version 20), 13 additional retracted papers were identified. These include 3 articles from MEDLINE, 4 articles from Web of Science, 2 articles from Scopus, and 4 articles from EMBASE. Therefore, 15, 768 articles were checked for duplicate entries.

In the second screening step, duplicate detection was performed on these 15, 768 articles. As a result, 13, 103 potential duplicates were identified, and 8, 148 duplicates were initially removed, leaving 7, 620 unique articles. However, after importing these 7, 620 articles into the Deduplicator of the Systematic Review Accelerator (SRA) tool [28], we identified and removed an additional 4 duplicates, resulting in a final total of 7, 616 unique articles remaining for the title and abstract screening step. As automation tools to resolve duplicates, we used EndNote [17], Rayyan.ai [16], and The Deduplicator of the Systematic Review Accelerator (SRA) [28] tool.

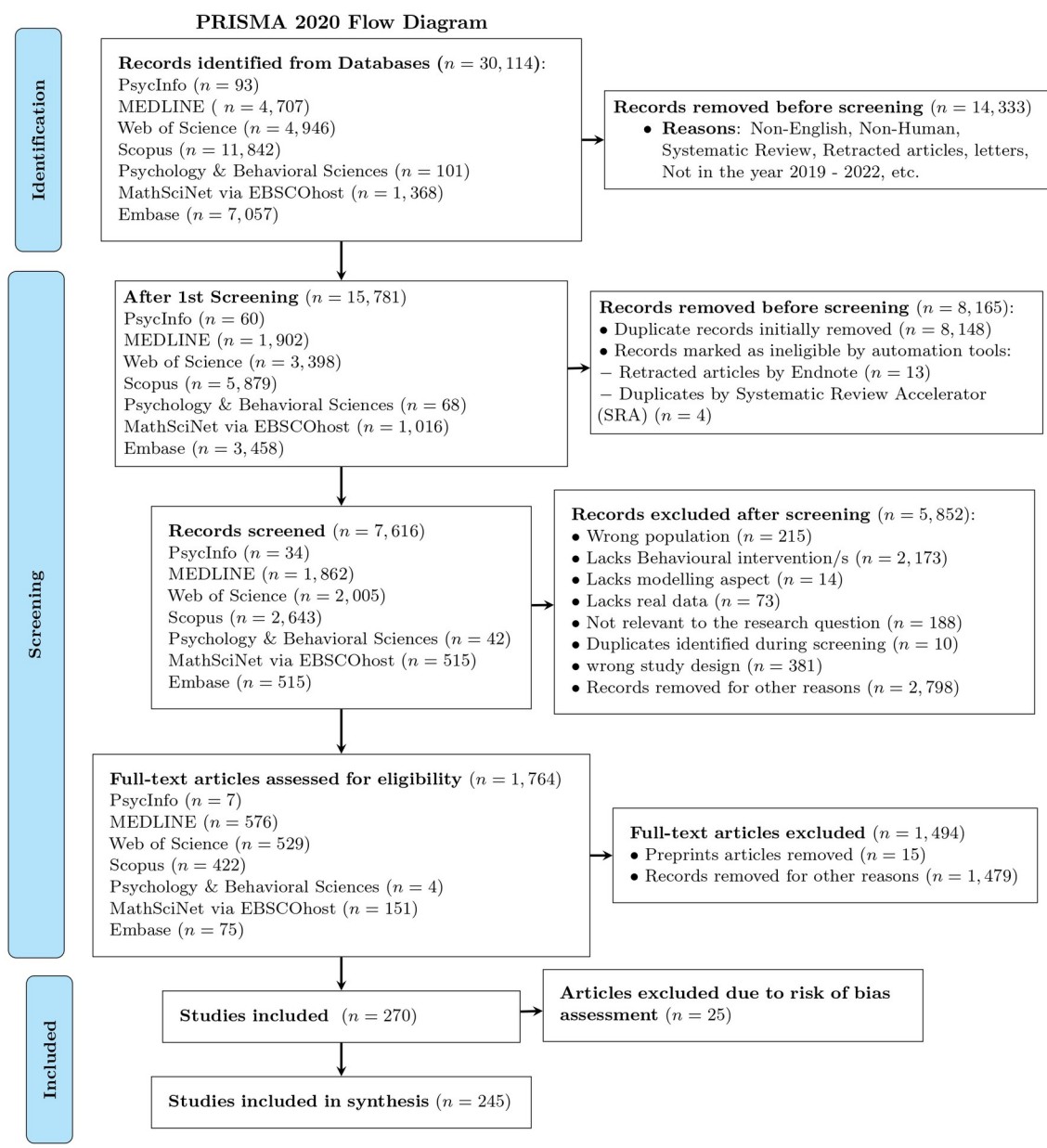

**Fig 1. PRISMA 2020 flow diagram for new systematic reviews which included searches of databases.** The diagram illustrates the stages of the systematic review process: identification of records from databases, removal of duplicates, screening for eligibility, and inclusion of studies in the final synthesis. Specific levels include: Level 1a and 1b for record identification and removal before screening, Level 2a and 2b for the first screening and further removals, Level 3a and 3b for records screened and excluded, Level 4a and 4b for full-text assessment and exclusions, and Levels 5a, 5b, and 6 for included studies and synthesis.

After the de-duplication process, we assessed 7,616 articles in the 'Title and Abstract' screening process, during which 5,852 were excluded, and the remaining 1,764 articles were screened for their full contents. We initially chose the databases that contained fewer articles for screening. Since 15 articles were still found as preprints, we excluded these and reviewed the full contents of the remaining 1,749 articles. During the full article screening process, we excluded 1,478 articles, leaving us with 270 remaining. Finally, we extracted the data items in

subsection Data items from the 270 articles included. As a summary, the process of the inclusion of articles is displayed using a flow diagram (See Fig 1).

During the full-text screening process, some studies that initially appeared to meet the inclusion criteria were excluded. For example, [29] titled "Analyzing the impact of the media campaign and rapid testing for COVID-19 as an optimal control problem in East Java, Indonesia" was excluded due to incorrect demographics data. The study inaccurately reported the population of East Java as 49, 316, 712, which is the population of West Java. This significant error would affect the validity of the SEIR model used in the study. We also assessed the relevance of the included articles to the research question and the validity of their methodologies. For example, one article by [30] was excluded from the study due to concerns regarding the realism of its application. Specifically, the researchers utilized parameters estimated from South Korea to model the COVID-19 epidemic in the United States, which raised doubts about the generalizability of their findings to the US context. Additionally, 25 articles were excluded from the Synthesis due to a high risk of bias identified in the risk assessment process. However, these articles were included in the descriptive statistics since they were still useful for summarizing general trends or characteristics, even if they were not considered reliable enough for the main synthesis [31–35].

## Study characteristics

This systematic review aims to synthesize the findings of studies that have modeled the impact of behavioral interventions during the COVID-19 pandemic [36]. The extracted data includes information on $270 - 25 = 245$ unique articles that examined the modeling of the impact of behavioral interventions during COVID-19 across different countries. These 245 articles generated 380 data entries due to the inclusion of multiple characteristics such as different countries, outcome measures, and key findings. The original 270 articles generated 406 data entries, but 26 of these had a high risk of bias assessment score.

We grouped the studies by their respective continents, as shown in Table 1. The majority of the studies were conducted in Asia (37.1%), followed by Europe (28.2%) and North America (19.2%). This distribution reflects the global impact of the COVID-19 pandemic and the extensive research efforts undertaken across different regions to understand and mitigate its spread through various behavioral interventions. The studies employed different modeling approaches, including compartmental models such as SEIR and SEIQR and logistic regression analysis to assess the effectiveness of interventions such as social distancing, mask-wearing, and self-protection measures. Most of (90.3%) the articles used compartmental models. Additionally, 62.6% studies were open-access, which is important during a pandemic because it helps share information quickly and widely.

We can also see in Table 1 that the number of studies has been decreasing over the years with the majority of the studies (41.3%) conducted in 2020. Most studies (94.5%) utilized secondary data, and 95% used "modeling and simulation" study design.

The word cloud in Fig 2 indicated a wide range of countries where COVID-19 studies were conducted. The US has the highest number of studies (57) in the dataset. China is the second most prominent, with 45 studies, showing considerable research interest and contributions. India, Brazil, and Italy also appear prominently, indicating substantial research activity in these countries. It seems that large populations and large national science budgets contribute to having more studies on this topic [37].

From the word cloud in Fig 3, the predominance of compartmental models becomes clear. The SEIR Model is the most frequently used model followed by the SIR model. The epidemiology model, logistic growth model, and compartmental model were also used, indicating that a

**Table 1. Descriptive statistics of studies by various characteristics.**

| No | Characteristic | Category | Freq. | Percent(%) |
|---|---|---|---|---|
| 1 | Continent | Africa | 13 | 3.4 |
| | | Asia | 141 | 37.1 |
| | | Australia (Oceania) | 4 | 1.1 |
| | | Europe | 107 | 28.2 |
| | | North America | 73 | 19.2 |
| | | South America | 30 | 7.9 |
| | | Worldwide | 12 | 3.2 |
| 2 | Population consideration | No | 55 | 14.5 |
| | | Yes | 325 | 85.5 |
| 3 | Open access | No | 142 | 37.4 |
| | | Yes | 238 | 62.6 |
| 4 | Compartmental | No | 37 | 9.7 |
| | | Yes | 343 | 90.3 |
| 5 | Type of data | Secondary data | 359 | 94.5 |
| | | Primary data | 13 | 3.4 |
| | | Experimental data | 4 | 1.1 |
| | | Mixed data | 4 | 1.1 |
| 6 | Study design | Modeling and simulation study | 361 | 95.0 |
| | | Predictive modeling study | 12 | 3.2 |
| | | Observational study | 7 | 1.8 |
| 7 | Publication year | 2022 | 102 | 26.8 |
| | | 2021 | 121 | 31.8 |
| | | 2020 | 157 | 41.3 |
| | **Total** | | 380 | 100% |

- "Population consideration" refers to whether the study addressed small settings or entire populations. "Compartmental model" indicates whether the study employed compartmental models. "Open access" indicates whether the article is accessible to everyone publicly.
- "Modeling and simulation study" refers to studies using simulation models, such as compartmental models (e.g., SEIR), to simulate disease progression over time. "Predictive modeling study" denotes studies using predictive methods, such as logistic regression, to forecast specific outcomes.

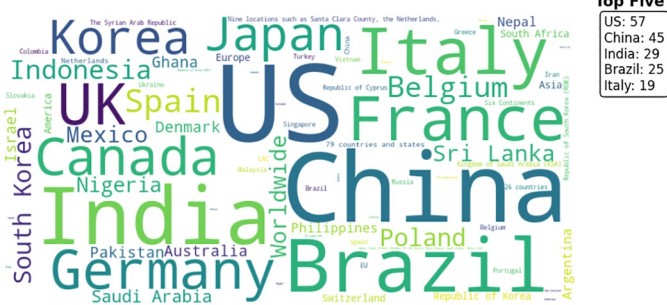

**Fig 2. Word cloud of countries considered in the study.** The figure shows a word cloud representing the frequency of countries used in the study. The size of each country's name corresponds to its frequency, with the most frequently mentioned countries appearing larger. The top five countries, based on their frequency, are listed in the upper right corner: US (57), China (45), India (29), Brazil (25), and Italy (19).

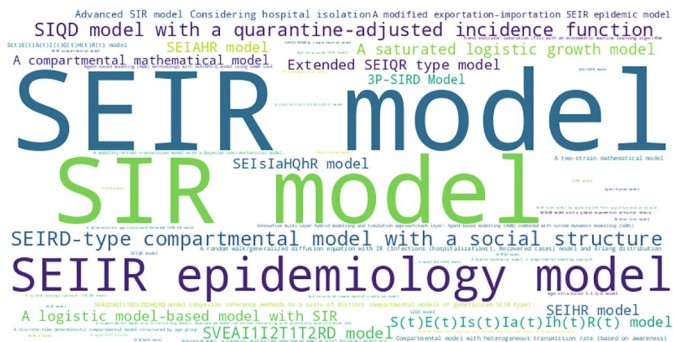

**Fig 3. Types of models used in studies included in the systematic review.** This word cloud depicts the frequency of different models employed in the studies included in the systematic review. The size of each model name reflects how commonly it was used across the studies. Prominent types, such as "SEIR model," "SIR model," and "SEIIR epidemiology model," stand out, indicating their prevalent use in the reviewed literature.

broad range of modeling techniques were employed to study the effects of behavioral interventions.

All studies varied widely regarding study design, population, interventions, and outcomes measured. Out of the 245 unique included articles, the key characteristics for the selected five articles are summarized in Table 2. The detailed characteristics of all studies are available in an Excel file, which can be accessed through the OSF link https://osf.io/dy9ck/.

## Risk of bias

Fig 4 displays a summary plot of the risk of bias assessment across 15 domains for all 270 articles. We can see that the overall risk of bias assessment indicates that most domains exhibit a low risk of bias, suggesting well-defined objectives and structures across the studies. Specifically, domains such as "Research Question, Goals, and Scope", "Model Structure and Assumptions", and "Data Informed Model" predominantly display low risk (green). However, areas like "Sensitivity and Stability Analyses" and "Transparency" show moderate risk (yellow), indicating some concerns. Bias-related domains-"Selection Bias", "Performance Bias", "Detection Bias", "Attrition Bias", and "Reporting Bias", have varying levels of risk, with occasional instances of high risk (red). Notably, critical concerns are observed in "Attrition Bias" and "Confounding" highlighting potential issues that may significantly affect the validity of findings.

Similarly, from Fig 5 and from S6 Table, the overall risk of bias assessment reveals that 62.6% of the studies included in the analysis had a low risk of bias, indicating strong methodological rigor and a high level of confidence in their findings. 24.1% of the studies were classified as having a moderate risk of bias, suggesting that they generally employed sound methods but may contain some elements that could introduce minor biases. A smaller fraction, 9.3%, of the studies exhibited a high risk of bias, indicating significant methodological concerns that could compromise the reliability of their results. Overall, the predominance of low and moderate-risk studies supports the robustness of the synthesized results, despite the presence of some studies with higher or unclear risks.

We assessed the risk of bias for the initially included 270 unique studies (E_1 to E_270) across 15 domains (D1 to D15). Fig 6 displays the risk of bias assessment for 85 out of the 270 articles, where the left panel includes studies that scored low risk, while the second figure highlights the 25 articles that scored high risk. The color-coded circles represent the level of bias:

**Table 2. Summary of extracted articles.**

| No | Authors | Study ID | Title | Year | Country | Model Name | Study Design |
|---|---|---|---|---|---|---|---|
| 1 | Ajbar et al. | https://doi.org/10.3855/jidc.13568 | Modelling the evolution of the coronavirus disease (COVID-19) in Saudi Arabia | 2021 | Saudi Arabia | SEaImIR-H | Modeling study |
| 2 | Brown | https://doi.org/10.1073/pnas.2105292118 | A simple model for control of COVID-19 infections on an urban campus | 2021 | US | Modified SEIR Model | Modeling study |
| 3 | Ritsema et al. | https://doi.org/10.2196/31099 | Factors Associated With Using the COVID-19 Mobile Contact-Tracing App Among Individuals Diagnosed With SARS-CoV-2 in Amsterdam, The Netherlands: Observational Study | 2022 | The Netherlands | Logistic Regression | Observational study |
| 4 | Al-Harbi and Al-Tuwairqi | https://doi-org.proxy-ub.rug.nl/10.1371/journal.pone.0265779 | Modeling the effect of lockdown and social distancing on the spread of COVID-19 in Saudi Arabia. | 2022 | Saudi Arabia | SEIR model | Modeling study |
| 5 | Alleman et al. | https://doi.org/10.1016/j.epidem.2021.100505 | Assessing the effects of non-pharmaceutical interventions on SARS-CoV-2 transmission in Belgium using an extended SEIQRD model and public mobility data. | 2021 | Belgium | Extended SEIQRD metapopulation model | Modeling study |

| Study ID | Sample Size | Type of Data | Target Population | Intervention Type | Outcome Measure |
|---|---|---|---|---|---|
| https://doi.org/10.3855/jidc.13568 | 34, 813, 871 | Secondary data | Individuals of Saudi Arabia | Self-protection measures such as social distancing and wearing masks | Number of COVID-19 cases |
| https://doi.org/10.1073/pnas.2105292118 | 10, 000 | Primary data | Undergraduate students at Boston University | Public health protocols, including surveillance testing, contact tracing, isolation, and quarantine | Predicted number of infections and detected cases within a university community |
| https://doi.org/10.2196/31099 | 29, 766 | Secondary data | Individuals diagnosed with SARS-CoV-2 in Amsterdam | Contact-tracing app usage | App usage among individuals diagnosed with SARS-CoV-2 |
| https://doi-org.proxy-ub.rug.nl/10.1371/journal.pone.0265779 | 34, 218, 169 | Secondary data | Individuals of Saudi Arabia | Lockdown and social distancing | Impact of lockdown and social distancing on COVID-19 spread |
| https://doi.org/10.1016/j.epidem.2021.100505 | 22, 136 | Secondary data | Patients in Belgian hospitals | Quarantine, Lockdown measures, school closures, and reduction in mobility | Hospitalization rates, mortality rates in hospitals or ICUs, the average time from symptom onset to hospitalization, Effective reproduction number (Re) |

| Study ID | Basic Reproduction Number | Effective Reproduction Number | Outcome Measure Results and Key Findings | Population Consideration (0 = n, 1 = N) | Compartmental (Yes = 1, No = 0) | Open Access (Yes = 1, No = 0) |
|---|---|---|---|---|---|---|
| https://doi.org/10.3855/jidc.13568 | 4.75 | $1.159 \times 10^{-08}$ | The total number of infected individuals would have reached a peak of 66, 750 by 4 May 2020 and the COVID-19 would have decreased in intensity by 99% by the end of June 2020; $R_0 = 4.75 > 1$, indicating rapid disease spread. The Basic Reproduction Number ($R_c$) for Scenario 1 is approximately 0.00000001159. | 1 | 1 | 1 |

*(Continued)*

**Table 2.** (Continued)

| | | | | | | |
|---|---|---|---|---|---|---|
| https://doi.org/10.1073/pnas.2105292118 | 2.5 | 0.48 | Surveillance testing twice per week can significantly offset transmission rates higher than reported for COVID-19. The model predicts approximately 20.7 cases within the university community at any time, with around 6.2 cases detected per day. | 0 | 1 | 1 |
| https://doi.org/10.2196/31099 | NA | NA | 16.2% of SARS-CoV-2 positive cases reported app use; Characteristics associated with app usage were identified through logistic regression analysis. | 1 | 0 | 1 |
| https://doi-org.proxy-ub.rug.nl/10.1371/journal.pone.0265779 | 1.8872 | 0.5242 | Experiment 1: $R_c = 0.5242$ indicating disease eradication. Experiment 2: $R_c = 1.8872$ indicating persistent disease presence. | 1 | 1 | 1 |
| https://doi.org/10.1016/j.epidem.2021.100505 | 4.16 | 0.62 | Average time from symptom onset to hospitalization: 6.4 days; Mortality in hospital: 21.4%, Mortality in ICU: 46.3%. | 0 | 1 | 1 |

green for low risk, yellow for moderate risk, and red for high risk, with '+' and '−' symbols indicating low and high risk, respectively. Most studies across most domains demonstrate a low risk of bias, as indicated by the prevalence of green circles. This suggests that, generally, the studies adhere to high methodological standards. Certain studies, like E_131 and E_44,

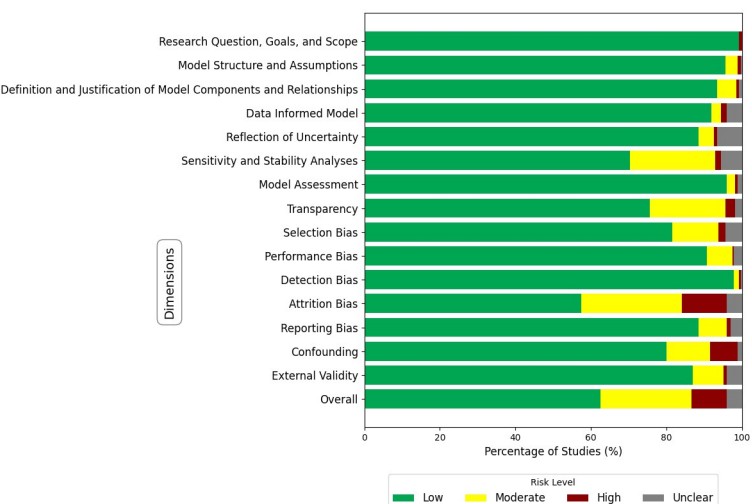

**Fig 4. A summary plot of the risk of bias assessment across 15 domains.** The figure summarizes the risk of bias across 15 domains and the "Overall" domain. D1: Research Question, Goals, and Scope. D2: Model Structure and Assumptions. D3: Definition and Justification of Model Components and Relationships. D4: Data-Informed Model. D5: Reflection of Uncertainty. D6: Sensitivity and Stability Analyses. D7: Model Assessment. D8: Transparency. D9: Selection Bias. D10: Performance Bias. D11: Detection Bias. D12: Attrition Bias. D13: Reporting Bias. D14: Confounding. D15: External Validity. D16: Overall.

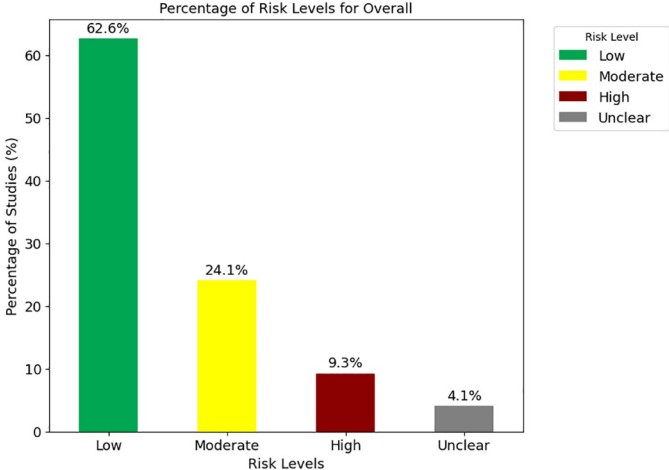

**Fig 5. Percentage of risk levels for the "Overall" risk of bias.** The figure illustrates the distribution of risk levels across studies. The percentage of studies categorized as Low risk is 62.6%, Moderate risk is 24.1%, High risk is 9.3%, and Unclear risk is 4.1%.

exhibit high risk in multiple domains. These studies appear to have significant methodological weaknesses that could impact the validity of their findings. The detailed risk of bias assessments for all studies is displayed in S2 and S3 Figs.

## Results of individual studies

Based on our research question "Under what conditions can behavioural interventions impact COVID-19", we reported the key findings in three formats: a descriptive analysis of $R_0$, a regression analysis of $R_0$, and a narrative analysis (a narrative approach to synthesis) of general outcome measures, including COVID-19 cases and deaths, transmission rates, and $R_0$. We selected 245 unique articles with 380 entries having low, moderate, and unclear risk of bias assessment scores and considered them for further analysis.

**Descriptive analysis of the reproduction number ($R_0$) as outcome measures.** From Table 3, we can see that the average $R_0$ value across the studies is approximately 3.184 with a standard deviation of 1.891, indicating that, on average, an infected individual could potentially spread the disease to about 3.184 other people in a completely susceptible population. From the sample size, the large standard deviation of 80, 500, 000, 000 reflects significant variation in population sizes studied, likely due to a mix of global, national, and regional focus areas.

Table 4 shows significant variation in mean and median $R_0$ values across continents. For instance, higher $R_0$ values observed in continents like Asia and South America might reflect higher transmission rates due to denser urban populations or variations in compliance with preventive measures. The standard deviation (SD) values also highlight the spread of $R_0$ estimates, suggesting that some regions within these continents may experience significantly different transmission dynamics.

Fig 7 displayed the boxplot of $R_0$ for continents. The median $R_0$ value is around 2.7 for Asia, with a relatively wide interquartile range (IQR), indicating variability in $R_0$ values. There are several outliers above 8, suggesting a few studies reported significantly higher $R_0$ values. The median $R_0$ value is about 2.9 for Europe, with a wide IQR, indicating greater variability across studies. There are few outliers and most studies in Europe report $R_0$ values between 0.5 and

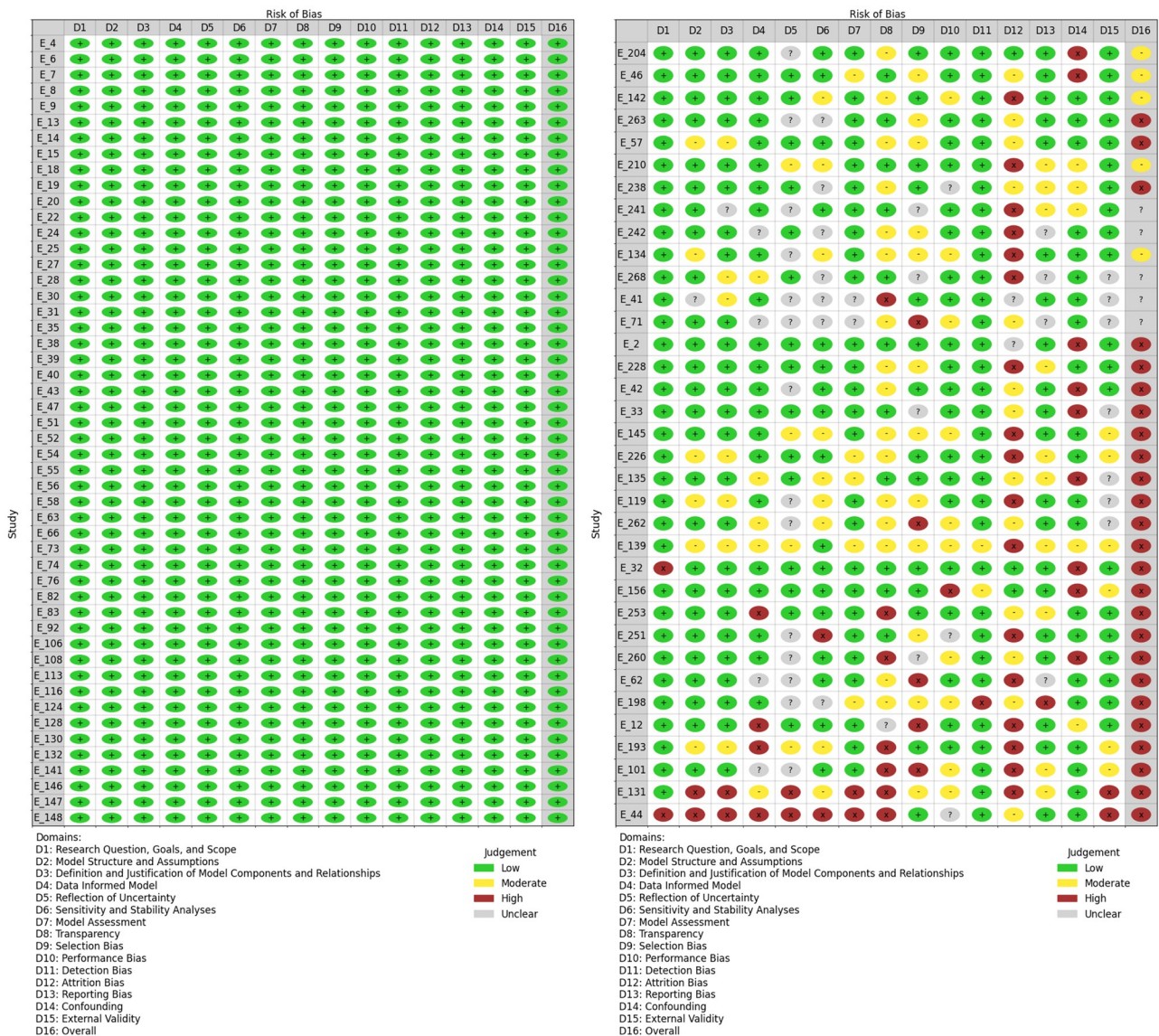

**Fig 6. Risk of bias judgments across studies: For the 50 studies with all low risk and the 35 studies including the 25 high-risk studies.** The figure provides a detailed overview of the risk of bias judgments for 85 out of the 270 articles. The left panel includes studies that scored low risk, while the right panel highlights the 25 articles that scored high risk. The colors indicate the level of risk: green for Low, yellow for Moderate, red for High, and gray for Unclear.

**Table 3. Descriptive statistics for $R_0$ and population size.**

| No. | Variable | Mean | Median | Std. Deviation | Min | Max |
|---|---|---|---|---|---|---|
| 1 | $R_0$ | 3.184 | 2.9 | 1.891 | 0.086 | 10.87 |
| 2 | **Population Size** | $4.86 \times 10^9$ | $2.65 \times 10^7$ | $8.05 \times 10^{10}$ | $2.88 \times 10^2$ | $1.39 \times 10^{12}$ |

**Table 4. Descriptive statistics for $R_0$ across different continents.**

| No | Continent | Mean | Median | SD | Min | Max |
|---|---|---|---|---|---|---|
| 1 | Africa | 2.29 | 2.01 | 0.87 | 1.02 | 4.09 |
| 2 | Asia | 3.27 | 2.68 | 1.99 | 0.0856 | 10.87 |
| 3 | Australia (Oceania) | 1.88 | 1.91 | 0.82 | 1.06 | 2.63 |
| 4 | Europe | 3.08 | 2.90 | 1.86 | 0.80 | 10.00 |
| 5 | North America | 3.33 | 3.00 | 1.88 | 1.03 | 9.40 |
| 6 | South America | 3.25 | 3.00 | 2.19 | 1.01 | 10.62 |
| 7 | Worldwide | 3.76 | 3.79 | 1.10 | 1.80 | 5.00 |

4.5. North America and South America have similar distributions of $R_0$ with a median $R_0$ value is around 3. There are outliers above 8 for North America, indicating some high $R_0$ estimates but South America has fewer outliers. The median is around 1.9 for Australia (Oceania) but around 2 for Africa. Australia (Oceania) has a lower central tendency compared to other continents.

**Regression analysis results for $R_0$.** Since a meta-analysis was not possible due to the heterogeneity of studies, we conducted a linear regression analysis to identify the relationship between the reported $R_0$ and various characteristics. The dependent variable was $R_0$ and the independent variables were population consideration (binary), compartmental (binary), type of data (categorical; with "Experimental data" as baseline level), sample size, Study design (categorical; with "modeling and simulation study" as baseline level), Continent (categorical; with "Africa" as baseline level), and Open access (binary).

From S4 Table, the inclusion of the variable sample size ($ni$) in the model did not significantly contribute to the predictive power for $R_0$, as indicated by its coefficient being near zero and its non-significant p-value. Removing the variable "sample size" improved the model's overall fit, yielding a significant F-statistic, suggesting that the remaining variables provide a better predictive structure for $R_0$. Additionally, based on the Variance inflation factor (VIF) values in S5 Table, multicollinearity is not a concern among the independent variables, affirming the robustness of the model without sample size.

Table 5 summarises the model coefficients in the regression analysis of $R_0$ without the sample size variable. The F-statistic ($F = 1.888$, $p = 0.027$) tests the null hypothesis that all

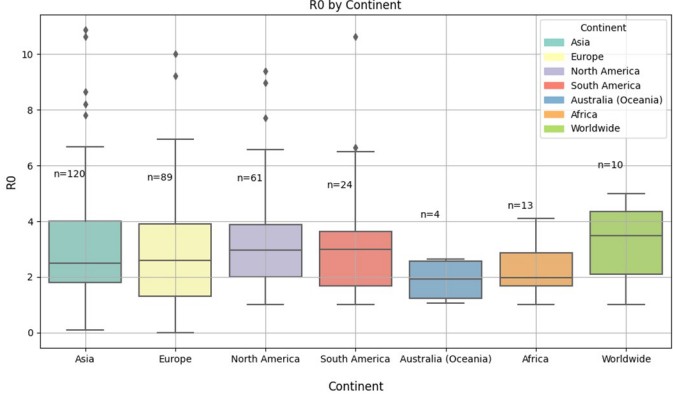

**Fig 7. Boxplot of $R_0$ by continent.** The figure illustrates the distribution of the basic reproduction number ($R_0$) across different continents using box plots. Each box plot represents the range of $R_0$ values for a specific continent, showing the median, interquartile range, and outliers.

**Table 5. Summary of model coefficients and 95% confidence intervals without the sample size (`ni`) variable.** $R^2$: 0.07951, Adjusted $R^2$: 0.03739, F-statistic: 1.888 ($df$ = 14, 306, p-value: 0.02711).

| No. | Variable | Category | Estimate | Std. Error | t value | Pr(> \|t\|) | 95% CI | |
|---|---|---|---|---|---|---|---|---|
| | | | | | | | 2.5% | 97.5% |
| 1 | (Intercept) | | 3.308 | 2.017 | 1.640 | 0.102 | −0.661 | 7.276 |
| 2 | Population Consideration | No (Ref.) | | | | | | |
| | | Yes | −0.155 | 0.331 | −0.466 | 0.641 | −0.807 | 0.498 |
| 3 | Compartmental | No (Ref.) | | | | | | |
| | | Yes | −0.304 | 0.517 | −0.588 | 0.557 | −1.321 | 0.713 |
| 4 | Type of data | Experimental data (Ref.) | | | | | | |
| | | Mixed data | −2.710 | 2.181 | −1.242 | 0.215 | −7.001 | 1.582 |
| | | Primary data | −1.419 | 2.154 | −0.659 | 0.510 | −5.658 | 2.820 |
| | | Secondary data | −1.249 | 1.904 | −0.656 | 0.512 | −4.996 | 2.497 |
| 5 | Study design | Modeling and simulation study (Ref.) | | | | | | |
| | | Observational study | −2.170 | 1.487 | −1.460 | 0.145 | −5.095 | 0.755 |
| | | Predictive modeling study | −1.540 | 0.823 | −1.870 | 0.062 | −3.160 | 0.080 |
| 6 | Continent | Africa (Ref.) | | | | | | |
| | | Asia | 0.997 | 0.549 | 1.817 | 0.070 | −0.083 | 2.076 |
| | | Australia (Oceania) | −0.155 | 1.067 | −0.145 | 0.885 | −2.254 | 1.945 |
| | | Europe | 0.865 | 0.561 | 1.541 | 0.124 | −0.239 | 1.970 |
| | | North America | 1.055 | 0.576 | 1.833 | 0.068 | −0.077 | 2.188 |
| | | South America | 1.228 | 0.656 | 1.871 | 0.062 | −0.064 | 2.519 |
| | | Worldwide | 1.395 | 0.803 | 1.738 | 0.083 | −0.185 | 2.975 |
| 7 | Open access | No (Ref.) | | | | | | |
| | | Yes | 0.786 | 0.227 | 3.456 | 0.001 * | 0.338 | 1.233 |

*Significant at 5% level.

– Ref. indicates for a reference category.

regression coefficients (excluding the intercept) are equal to zero. The result indicates that the overall model provides a statistically significant improvement in explaining the variation in $R_0$ compared to a model with no predictors. However, the model explained only 8% of the variance in $R_0$ ($R^2$ = 0.080), with an adjusted $R^2$ of 0.037, indicating limited explanatory power.

The variable "Open access" was highly significant ($p < 0.001$), with a positive coefficient of 0.786, suggesting that open-access studies are associated with higher $R_0$ values. For the Continent variable, Asia, North America, South America, and Worldwide had positive coefficients with marginal significance at a 10% level ($p$-values ranging from 0.062 to 0.083), indicating higher $R_0$ values compared to the reference continent, Africa. The Study design variable showed that predictive modeling studies were associated with lower $R_0$ values compared to the reference category "modeling and simulation study", but these findings were only significant at the 10% level ($p = 0.062$).

**Narrative analysis.**

1. **The basic reproduction number** ($R_0$): Studies from various countries (Belgium, Germany, the US, and Saudi Arabia) showed that behavioural interventions helped reduce the $R_0$ (basic reproduction number) below 1, effectively controlling the spread of the virus [32, 38–41]. A study on the sample of the Iraqi population of size 5.000 by [42] indicated that the curfew and social distancing measures can reduce the basic reproduction number ($R_0$) to below 1, preventing outbreaks. Increasing media coverage will not completely prevent

outbreaks, but can reduce transmission by increasing awareness. Reducing $R_0$ is key to controlling the disease. Reducing social distance leads to an increase in the outbreak of the disease. Increasing the probability of an individual's response to the curfew leads to lowering the $R_0$ below unity and subsequently controlling the spread of COVID-19 [42]. When the quarantine ratio is greater than 65%, the reproduction number ($R_0$) can be below unity [43]. A study in Indonesia by [44] indicated that Without vaccination intervention, the transmission rate $\beta$ must be reduced by greater than 39% to maintain $R_0$ less than unity, and thus provide an opportunity to eliminate COVID-19 from the population. A study by [45] indicated that a range of values of $(0.1 - 0.2)$ for both $\beta$ and $\eta_0$ will peak the control reproduction number, $R_c$ in the range $(0.1 - 0.88)$ for Ghana and $(0.2 - 0.95)$ for Egypt. They said that the rate of quarantine through doubling enhanced contact tracing, adhering to physical distancing, adhering to wearing of nose masks, sanitizing-washing hands, and media education remains the most effective measures in reducing $R_0$ to less than unity. The researchers [46] indicated that banning $\geq 50$ gatherings was sufficiently effective to decrease, $R_0$ in some locations; including most New York counties; Massachusetts counties; and Prince George's, Maryland. However, for other counties, the drop of $R_0$ was significant only after issuing the stay-at-home order.

2. **The timing of interventions**: The timing and strictness of interventions played a crucial role. A study by Matrajt and Leung found that implementing interventions 50 days after the first case resulted in delayed epidemic peaks with minimal peak reduction compared to earlier interventions [47]. A study by [36] found that lockdowns resulted in an 80.31% reduction in effective contacts (95% CI: 79.76−80.85%) and a subsequent decrease in reported COVID-19 cases during the initial two weeks of implementation. Even though the study by [48] doesn't specify a precise timeframe for the stages C/D, they use stages C/D to represent a later phase in the pandemic without vaccination. They found that without vaccination, the pandemic would slow down in a later stage C/D (Without vaccination, without lockdown/With vaccination, without lockdown) and even worse on the earlier release of the lockdown. The study entitled "A data-driven epidemic model to analyze the lockdown effect and predict the course of COVID-19 progress in India", [49] found that the infection rate decreased to 3 times lower than the initial rate after 6 weeks of lockdown. The peak and end of the epidemic were estimated in July 2020 and March 2021, respectively. Active infected cases at peak time could reach around 2 lakhs (200, 000 cases), with total infected cases potentially exceeding 19 lakhs (1, 900, 000 cases). A study by [50] indicated that the early lockdown in mid-March 2020 significantly helped in controlling the spread of COVID-19 despite the low number of initial cases in Ukraine.

3. **Intervention intensity/strictness**: Reducing adult contacts by 95% starting at day 50 significantly mitigated the epidemic peak, nearly eliminating cases. Lower reduction levels (25% and 75%) had less impact on peak reduction [23]. A study by [51] reported that if at least 80% of people wear a mask, even if only 40% effective, transmission on campus will likely not result in any deaths. Particularly, they said that the results are not very sensitive to the changes of the $R_0$ values (varying $R_0$ values of 1.8, 2.5, and 3) if 80% (widespread mask usage) of the population wears masks; indicating the control of the outbreak with 5, 9, and 13 cases respectively, and zero deaths. They also reported that there was a dramatic reduction in cases and almost no deaths if at least 40% of people were wearing a mask.

4. **Targeting age groups**: Reducing contacts of adults over 60 by 95% prevented 50% of cases in this group, 30% of hospitalizations, and 39% of deaths in all age groups. A 95% reduction in adult contacts under 60 caused a 98% drop in peak cases. A 75% reduction in adult

contacts under 60 resulted in a 91% decrease in peak cases. Adding child contact reduction further decreased hospitalizations by over 64% in all age groups (overall). The intervention of reducing contacts of all age groups by 25% led to a 69% reduction in cases at the epidemic peak. After the social distancing measures were lifted, the number of cases started to increase again for all intervention strategies except for the one targeting only adults over 60 due to their smaller population size and fewer contacts. The study by [47] highlighted significant uncertainty in the effectiveness of milder interventions (e.g., 25% contact reduction). Surprisingly, cases, and hence hospitalizations and deaths, can be reduced by 90% during the first 100 days if all groups reduce their contact with others, even when adults do so by only 25%. When only 25% of adults < 60 changed their contact habits, all interventions rebounded as soon as the intervention was lifted. A study in France by [52] indicated that weak populations, people in the age group over 60 years, have a high probability of dying. They also said the strong population has a considerably shorter confinement. As indicated in Wuhan by [53], children and adolescents were less susceptible to infection, but more infectious once infected, than individuals aged 20 years or older. Children's higher infectivity was affected by household size. They also found a higher susceptibility of infants (aged 0–1 years) to infection than older children ($\geq$ 2 years of age). Although children and adolescents were much less likely to have severe disease, they were as likely as adults to develop symptoms. Similarly, a study by [50] indicated that higher mortality was observed in the 50–70 years age group, with significant deaths even among the 40–50 age group, potentially due to co-morbidities. Children and young adults under 20 accounted for around 10% of cases, with some deaths, highlighting the importance of maintaining social distancing in nurseries and schools [52].

5. **Impact of combined intervention strategies**: A study by [54] concluded that the combination of three controls is more influential when compared with the combination of two controls and a single control. A study by [48] shows a higher peak in daily new cases without lockdowns (scenario C) compared to scenarios with lockdowns (A and B). The simulations showed that lockdowns helped decrease the transmission rate, highlighting the potential benefits of combining lockdown measures with other interventions. In contrast, a data-driven assessment by [55] in China highlighted the limitations of travel restrictions as a standalone measure. Although travel restrictions may be effective in the short term, they cannot eradicate the disease due to the risk of spreading to other regions. In all scenarios considered, the majority of cases remained contained within Hubei province, regardless of the travel restrictions. These restrictions had a relatively small impact on the temporal evolution of the disease in the rest of the country.

Mass gatherings like the one in China reported by [56] could have been a potential infection risk without any preventive strategies. However, the combined use of vaccination, nucleic acid testing, and face mask-wearing effectively protected the people against infection. They found that the use of any two of these strategies could significantly lower the infection rates [57].

A combination of media campaigns and rapid testing reduces the number of infected individuals significantly. The more aware the community is, the more readily the infection rate will decrease [29].

6. **Most effective type/s of intervention/s** NPIs like social distancing, mask-wearing, lockdowns, and school closures significantly impacted transmission rates.

- **Mask usage** had a substantial effect on reducing transmission. Studies from the US and Mississippi particularly highlight the significant drop in cases and deaths when a high

percentage of the population wore masks consistently [51]. They reported that at 20% mask use, the cases and deaths are over halved from 9314 cases and 37 deaths (without masks) to 4094 cases and 12 deaths (20% mask use). Another article titled "Optimal Control on COVID-19 Eradication Program in Indonesia under the effect of community awareness", found that medical masks have the greatest effect on determining the number of new infections [58].

- **Social distancing**: A study by [59] concluded that social distancing is the main nonpharmaceutical measure to end the novel coronavirus. A study by [60] found that among the measures, including social distancing in adults, spring semester postponement, diagnostic testing, and contact tracing; social distancing in adults showed the highest effectiveness. From a study by [61], the most effective double control strategy is isolation combined with detection. Maximum detection must occur at the beginning to ensure that infected individuals are rapidly transferred to hospitals and isolated for treatment as quickly as possible. The intensity of testing remained high until day 11, after which it gradually decreased to zero. This strategy effectively reduced the source of COVID-19 transmission within the population.

- **Awareness**: Community awareness plays a crucial role in determining the success of COVID-19 eradication programs [58]. A study by [62] in Wuhan reported that public health efforts promoted infection-prevention actions like mask-wearing, hand hygiene, and reduced mobility. They analyzed the influence of two key factors: sensitivity to payoff gain ($\kappa$), representing public awareness and willingness to take these preventive actions, and control measure effectiveness ($\alpha$). Their findings suggest that a higher $\kappa$ and a lower $\alpha$ (more effective control measures) can significantly reduce the COVID-19 outbreak size.

7. **Individual compliance and social factors influencing NPIs**: Individual compliance with NPIs significantly influenced the effectiveness of NPIs. Studies like the one in the Netherlands by [39] showed a lower uptake of contact tracing apps, potentially hindering their effectiveness. Behavioural interventions like physical distancing rely on individual compliance for effectiveness. Studies like [31] highlight the influence of social factors on adherence. Their research found that despite restrictions, nearly 10% of participants aged 18–59 reported intimate physical contact outside their household in the past month. This finding exemplifies how social needs like intimacy can influence compliance with behavioural interventions like physical distancing.

The findings indicate that the implementation of strong control measures, including government actions and mild self-protection measures, can significantly reduce the transmission rate of COVID-19. For instance, A study by [38] found that with such measures, the basic reproduction number ($R_0$) could be reduced to less than 0.001 suggesting the virus' spread would be effectively controlled. Similarly, a study by [40] found that when lockdown and social distancing measures were moderately applied ($\rho = 0.5$, $SD = 0.75$), the basic reproduction number ($R_0$) was reduced to 0.524 from 1.887, indicating that the community would be free of COVID-19. Another study by [63] demonstrated that a 20% increase in the effectiveness of lockdown measures combined with a 20% increase in face mask compliance will result in a 92% reduction in the cumulative number of deaths.

However, the effectiveness of non-pharmaceutical interventions appears to vary greatly across different settings and populations. Some studies observed that the number of cases dropped by more than 90% due to the implementation of continuous social distancing measures [64]. Similarly, other studies found that the complete lockdown contributed to a reduction in the effective contact rate by a factor of approximately 6.1, indicating successful

containment measures [65]. In contrast, others highlighted the importance of considering population-level factors, such as the size of the setting (small vs. large) [66–69].

Furthermore, although these interventions have a beneficial effect on reducing the reproduction rate, they have negative economic, social, and other consequences [64]. These consequences have hardly been studied, and are therefore also not considered in this systematic review.

## Results of syntheses

To provide an overview of the primary focus areas in the literature, we generated a word cloud from the keywords for inclusion. This visualization highlights the most frequently discussed themes and interventions, giving a clear picture of the main topics covered in the research on behavioral interventions during COVID-19 (See Fig 8).

The word cloud in Fig 8 reveals that "distancing" is the most frequently mentioned keyword, followed by "quarantine", and then by "Isolation and lockdown". It also highlights that most studies use compartmental models, particularly SEIR, to analyze and predict the effectiveness of these behavioral interventions. Keywords such as "Non-Pharmaceutical Interventions (NPIs)", "masks", "hand hygiene", and "contact tracing" show the range of non-pharmaceutical strategies used to mitigate pandemic impacts.

We generated a network visualization of the terms from the titles and abstracts. This visualization highlights the central themes and their interrelationships in the literature on behavioral interventions during COVID-19 [24]. The network visualization represents the co-occurrence of terms from the titles and abstracts of included studies, illustrating how different concepts

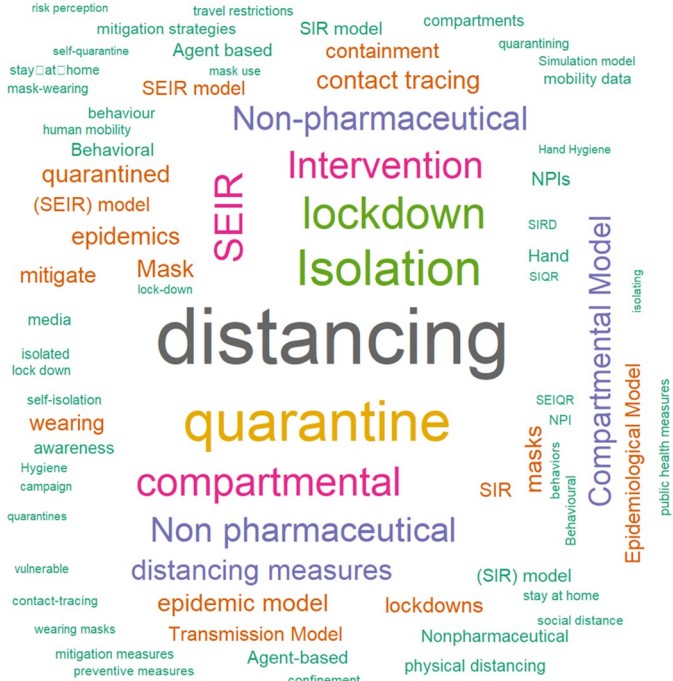

**Fig 8. Word cloud of keywords for inclusion.** The figure presents a word cloud generated from keywords used for inclusion criteria of the systematic review. The size of each keyword reflects its frequency or emphasis, with larger words indicating higher prominence. The most frequent key terms include "distancing," "quarantine," "isolation," "lockdown," and "compartmental models" such as SEIR.

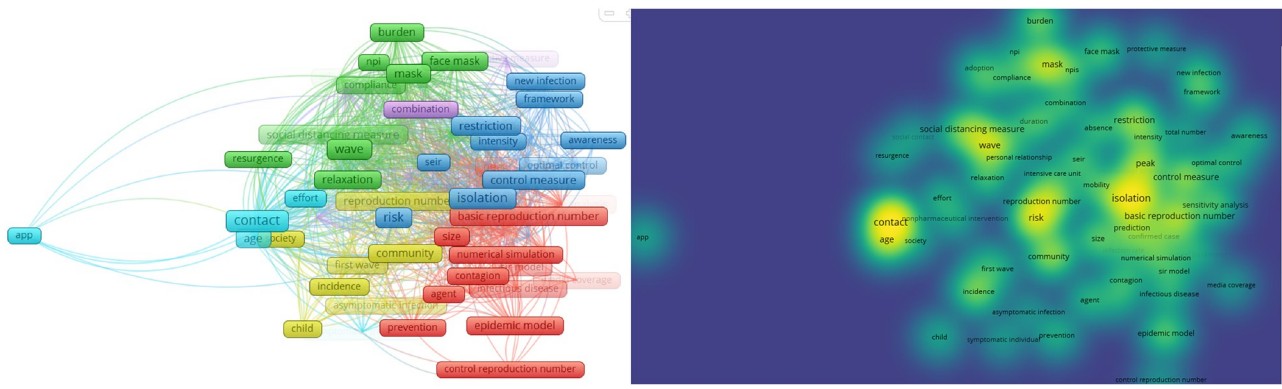

**Fig 9. Network and density visualization of terms from titles and abstracts.** The figure consists of two panels. The left panel shows a network visualization of terms from the titles and abstracts, with clusters representing related concepts in the literature. The right panel displays a density visualization, highlighting the frequency and centrality of key terms.

are related in the context of modeling the impact of behavioral interventions during COVID-19.

From the left panel of Fig 9, the green cluster includes terms such as "social distancing", "mask", and "npi", highlighting general non-pharmaceutical interventions (NPIs) aimed at preventing the spread of COVID-19. The blue cluster features terms such as "isolation", "SEIR", and "restriction", focusing on specific control measures and epidemiological modeling approaches. The red cluster encompasses terms like "epidemic model", "basic reproduction number", and "numerical simulation", indicating a strong emphasis on mathematical modeling and simulation studies. Finally, the yellow cluster includes terms like "community", "incidence", and "child", reflecting research on community-level interventions and demographic considerations. The term "app" is separate from the main clusters, suggesting a unique but significant interest in digital solutions for pandemic management, such as contact tracing applications or health monitoring tools.

Similarly, the density visualization in the right panel of Fig 9 offers a heatmap representation of term frequencies and their associations. We can see that the high-density areas (bright yellow to green) such as "contact", "age", "mask", "social distancing", "wave", "risk", and "isolation" indicate that these terms are central to the discourse on COVID-19 measures. Moderate-density areas (green to yellow) include terms like "basic reproduction number", "restriction", "peak", "control measure", "community", "incidence", and "burden", showing significant but less central discussions. Low-density areas (green to blue) such as "app", "child", "agent", "media coverage", "awareness", "protective measure", and "prevention" indicate less frequent discussions. This visualization provides a clear picture of the focus areas within the systematic review.

From the network visualization in Fig 10, the term "face mask" is linked with terms such as "contact", "testing", "control strategy", and "behavior". Similarly, the term "self isolation" is linked with terms including "testing", "contact", "contact education", "behavior", "awareness", and "control strategy". From the bottom panel of Fig 10, the term "media campaign" is linked with "awareness", "test", and "infected individual". Additionally, the term "physical distancing measure" is linked with terms including "adherence", "control strategy", "epidemic peak", "infection risk", "social contact", "self isolation", "testing strategy", and "surveillance".

From Fig 11, "travel" is linked with terms including "face mask", "behavior", "movement", "testing", "importation", and "adherence". Similarly, "travel ban" is linked to "importation",

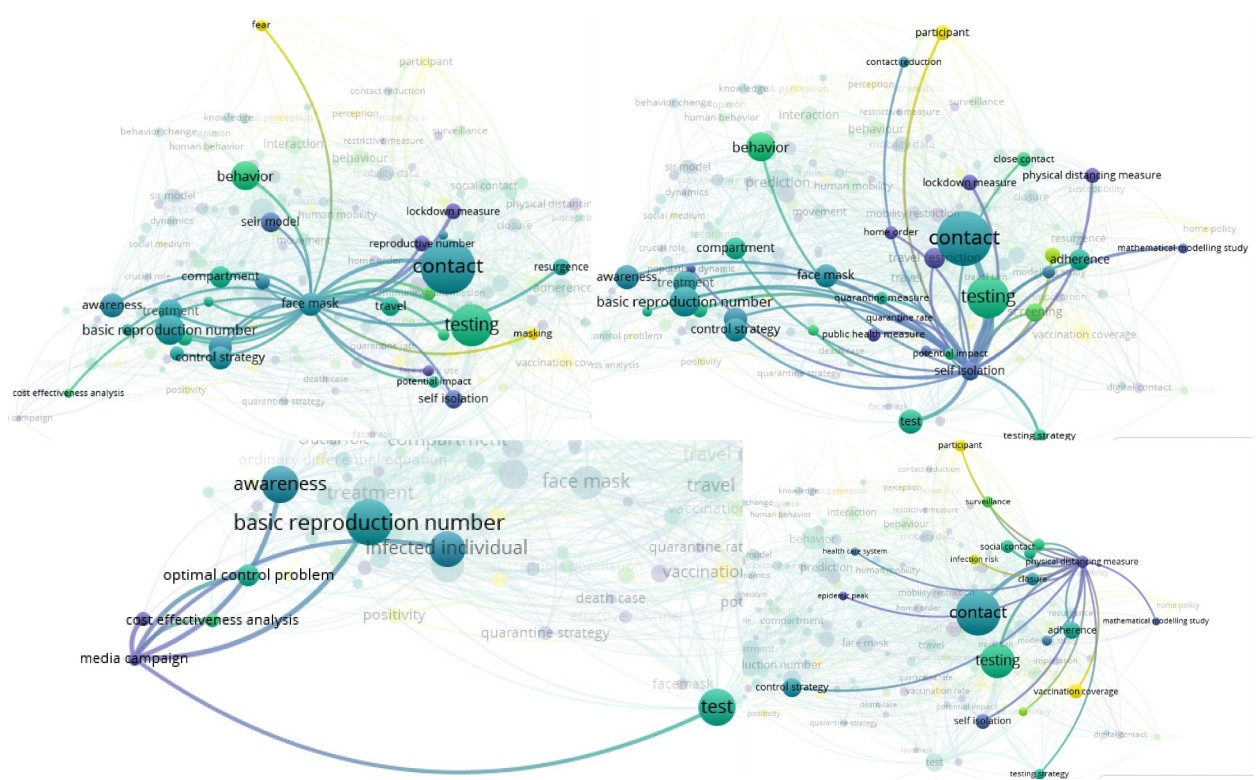

**Fig 10. Network visualization that shows terms linked with "face mask", "self isolation", "media", and "physical distancing measure".** The figure shows a network visualization of terms associated with "face mask", "self isolation", "media campaign", and "physical distancing measure", highlighting their connections to related concepts in the literature.

"movement", and "travel restriction". The connections illustrate how these concepts are linked in the context of COVID-19 measures and interventions. Additional network visualizations illustrating the relationships between terms such as "SEIR model", "app," and "preventive behavior" are provided in S4 and S5 Figs.

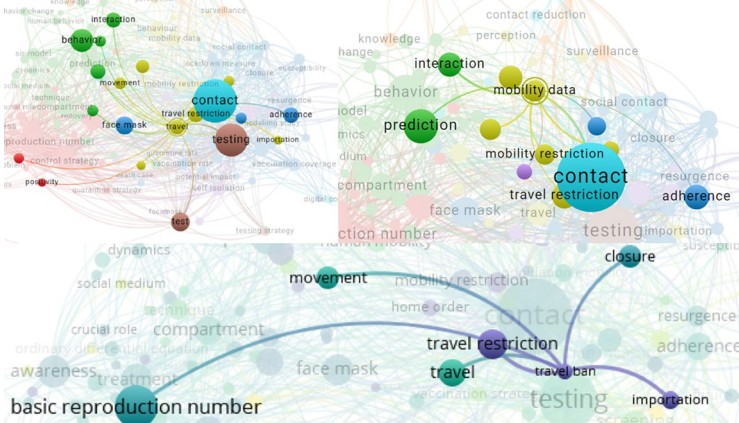

**Fig 11. Network visualization of terms related to travel and mobility restrictions.** The figure displays a network visualization showing the relationships between terms. The upper panel shows terms linked with "travel" or "mobility data" and the bottom panel shows the terms linked with the term "travel ban".

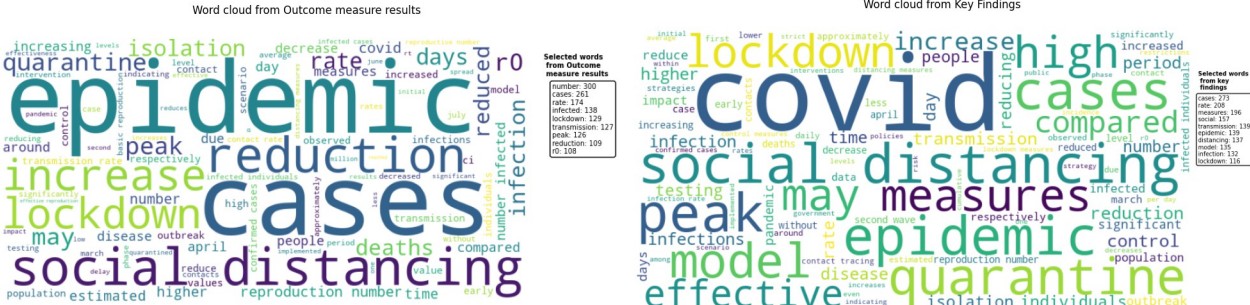

**Fig 12. Word clouds depicting the main outcome measure results and key findings characteristics from the systematic review.** The figure presents two-word clouds. The left panel shows the word cloud of outcome measure results, highlighting frequently analyzed terms such as "cases," "infection," "deaths," "reduction," and "peak." The right panel displays the word cloud for key findings, emphasizing terms like "social distancing," "quarantine," "lockdown," and "COVID".

**General outcome measures results.** Fig 12 represents the outcome measure results and key findings characteristics of the extracted textual data. Words such as "cases", "infection", "deaths", "reduction", "increase", $R_0$, and "peak" suggest that the studies frequently analyzed changes in infection rates, the effectiveness of interventions, and the impact on mortality rates.

Words such as "days", "time" and names of specific months imply that the outcomes were analyzed over specific time frames, reflecting the temporal dynamics of the pandemic and the interventions' effects. Terms such as "quarantine", "isolation" and "social distancing" indicate that the studies extensively discussed various behavioral and control measures implemented to curb the spread of COVID-19.

**Heterogeneity assessment.** Fig 7 in Results of individual studies subsection, illustrates a box plot of the basic reproduction number ($R_0$) across different continents, summarizing the variation in reported values from the included studies. The box plot reveals significant heterogeneity in $R_0$ estimates, with Europe and Asia showing broader ranges and higher outliers compared to regions like Africa and Australia (Oceania). This variability can be attributed to several factors, including differences in public health policies, sample size, and underlying population characteristics [70].

**Sensitivity analyses.** In the absence of a feasible meta-analysis due to high heterogeneity, in our primary analysis, we excluded articles assessed as having a high risk of bias, allowing us to consider studies with low, medium, and unclear risk of bias. For the sensitivity analysis, however, we focused exclusively on studies with a low risk of bias to assess the robustness of our findings and determine whether excluding studies with potential biases would alter the overall results. These analyses focused on the impact of varying inclusion criteria and the influence of outliers on the synthesized results.

The box plots in Fig 13 illustrate the distribution of $R_0$ values across different datasets. The original dataset includes studies before the risk of bias assessment, i.e., with a low, medium, unclear, and high risk of bias, whereas the filtered dataset includes only studies with a low risk of bias. The median $R_0$ is around 2.5 for the original dataset, with a significant number of studies reporting values above this range whereas the respective median $R_0$ value for the filtered dataset is around 3, which is higher, suggesting that higher-quality studies report slightly higher central estimates. For the original dataset, there is considerable variability, with $R_0$ values ranging up to approximately 16. Whereas, for the filtered dataset, the variability in $R_0$ values is reduced, with fewer outliers compared to the original dataset. The maximum $R_0$ value in this set is approximately 11, indicating a more consistent range of estimates.

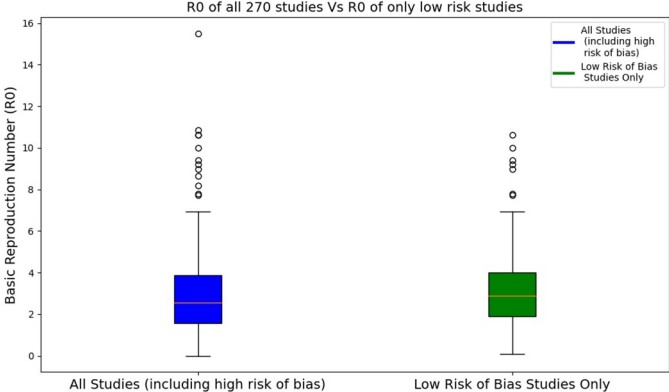

**Fig 13. Box plots of $R_0$ of all 270 articles versus $R_0$ of articles with only low risk of bias.** The left panel shows the $R_0$ values from the original dataset including articles with a high risk of bias, while the right panel shows the $R_0$ values of low risk of bias articles only.

Additionally, we plotted box plots that illustrate the distribution of $R_0$ values across different continents for both datasets. The box plots in Fig 14 indicate that the filtered dataset, which includes studies with only a low risk of bias, generally reports higher median $R_0$ values across most continents compared to the original dataset. Outliers are present in several continents in both datasets, particularly in North America and Europe. Across most continents, the filtered dataset reports higher median $R_0$ values, suggesting that the exclusion of studies with a higher risk of bias might eliminate studies with potentially lower $R_0$ estimates. In Africa, similar median values and variability were reported, indicating stable estimates across both datasets.

## Reporting biases results

In this subsection, our assessment focused on evaluating the completeness and transparency of outcome reporting. As we can see from Fig 4 of Risk of bias subsection, the reporting bias is included as one of the key domains in the risk of bias assessment criteria. While many studies

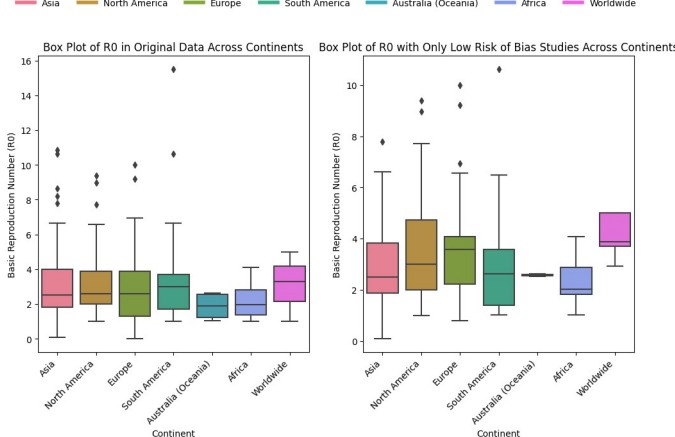

**Fig 14. Box plots of $R_0$ across continents.** The left panel shows the $R_0$ values from the original dataset across different continents, which includes studies with varying bias levels (low, medium, unclear, and high). The right panel presents the $R_0$ values from the filtered dataset, consisting only of studies with a low risk of bias.

are categorized as having a low risk of reporting bias, there are also a notable number of studies with moderate to high risk.

From S1 Fig and S6 Table, we can see that the majority (88.5%) of the studies were categorized as having a low-risk of reporting bias, indicating a high level of transparency and completeness in reporting outcomes for the majority of studies. However, there are 7.4% of studies with moderate risk, 3.0% with unclear risk, and 1.1% with high risk, highlighting some concerns regarding potential selective reporting or insufficient detail in some cases. These findings underscore the importance of improving reporting standards and ensuring comprehensive outcome reporting in future research.

## Certainty of evidence

- **Comprehensive search strategy**: We conducted a systematic literature search with a broad search strategy to identify relevant studies.

- **Transparent reporting**: We have reported the characteristics, methodologies, and key findings of each included study in detail.

- **Exploration of heterogeneity**: We explored potential sources of variation in the results using subgroup analyses.

- **Addressing potential bias**: We discussed potential limitations and sources of bias in the review, including publication bias.

## Discussion and conclusion

The systematic review examined the literature on the effectiveness of behavioral interventions, such as distancing, quarantine, and face mask-wearing, in mitigating the impacts of COVID-19. Due to the extreme heterogeneity (by design) in studies, a formal meta-analysis was infeasible, but our systematic and narrative review does give information about common findings, methodological approaches, and shortcomings of the studies under consideration.

### Methodological approach

The review also highlighted that the majority of studies utilized compartmental models, particularly the SEIR model, to analyze and predict the effectiveness of these behavioral interventions. This can only partially be explained by the fact that "compartmental model" was one of the explicit search terms in the literature search; as we also explicitly searched for other models. The widespread use of SEIR models reflects their ability to capture key dynamics of disease transmission and intervention impacts. These models effectively simulate epidemic progression across different scenarios, offering valuable insights that inform public health policies [7, 71].

### Common results

Our systematic review aligns with findings from previous studies. The word cloud analysis revealed that "distancing" was the most frequently mentioned keyword, followed by "quarantine" and "isolation and lockdown". These results were supported by [72], where the researchers measured the extent to which social distancing succeeds in reducing the contact rates of individuals. Another research by [73] also found that lack of social distancing and limited stay-at-home restrictions can impact COVID-19 spread, which aligned with our results. The

network visualization of the terms from the titles and abstracts further illustrated the central themes and their interrelationships in the literature on behavioral interventions. The network highlighted the connections between concepts like "non-pharmaceutical interventions", "masks", "hand hygiene", and "contact tracing", which represent the diverse range of non-pharmaceutical strategies used to mitigate COVID-19 impacts. These results were supported by many researchers [71, 73, 74].

The results of our systematic review identified that with early and stricter interventions, the transmission rates of COVID-19 are reduced. The significant reduction in COVID-19 cases associated with the early implementation of lockdowns is consistent with the findings of previous studies highlighting the timing of interventions as a critical factor in mitigating the effects of pandemics. Early action is crucial to flatten the epidemic curve and prevent overburdening the healthcare system [75, 76].

The variability in $R_0$ values across different continents, and years, in our review is consistent with other global studies that have highlighted strong regional differences in transmission dynamics and public health responses. This underlines the importance of tailoring public health strategies to the local context [74, 77–79].

As noted in our synthesis, the observed effectiveness of combined intervention strategies is supported by other studies encouraging a versatile approach to pandemic management. The synergistic effects of combining measures such as lockdowns, widespread mask use, and contact tracing are more effective than individual interventions alone [76, 80].

As older adults, particularly those over 60, are at a significantly higher risk of severe COVID-19, interventions targeting this age demographic deserve special attention. Our study suggests that interventions targeting older adults led to a 30% reduction in hospitalizations and a 39% decrease in overall deaths [81, 82].

The extensive analysis and inclusion of 245 studies following the risk of bias assessment provided a broad understanding of the impact of behavioral interventions during the COVID-19 pandemic. With 62.6% of studies classified as low risk of bias, our findings rest on a strong methodological foundation, providing confidence to the synthesized results. Meanwhile, 24.1% of studies that fell into the moderate risk category indicate minor methodological concerns that may slightly affect the validity of their findings. The 9.3% studies with a high risk of bias highlight areas where methodological approaches need to be strengthened, raising significant concerns about the reliability and validity of their conclusions.

During the systematic review process, we observed that some studies did not clearly explain how their data sources were selected or whether these sources were representative of the general population, leading to their exclusion from our review. Additionally, some studies normalized the sample size to 1, complicating the interpretation of specific population data. For instance, in our extracted data, 1.1% (3 out of 270) unique articles normalized the sample size to 1, and 20% (54 out of 270) unique articles did not specify the sample size.

In some cases, authors placed their data sources in the supporting material without explanation in the body of the paper, making it difficult for readers to ascertain if and how these data were used. For example, some studies using compartmental models included the population size ($N$) in supplementary materials instead of specifying it in their main reports, making it challenging to assess whether population size was considered if it was not clearly stated in the main text.

Similarly, some studies did not mention the basic reproduction number ($R_0$), a fundamental outcome measure in compartmental modeling studies, while others omitted the confidence interval of $R_0$, which impacts the ability to assess the precision and reliability of estimates. In our extracted data, 20.37% (55 out of 270) unique articles did not specify $R_0$.

We also noted selective reporting of outcomes, with some studies focusing only on positive findings while neglecting negative results. Additionally, some studies cited the models they used without explaining the model structure and assumptions, which delayed the ability to fully understand and replicate the study findings, resulting in being assigned a high risk of bias for the "Definition and Justification of Model Components and Relationships" domain. For example, in our extracted data, only 0.741% (2 out of 270) articles have a ''high'' risk of bias for this domain.

The absence of sensitivity analysis in some studies limits the robustness of the assessment of model outcomes to changes in key parameters. In our data, 22.593%(61) articles were rated as having 'moderate' risk, 1.481%(4) as "high" risk, and 5.555%(15) as "unclear" in the "Sensitivity and Stability Analyses" domain. Moreover, some studies combined the effects of behavioral interventions with vaccination, introducing confounding factors that complicate the interpretation of behavioral measures. In the "Confounding" domain, 80%(216) studies were rated "low" risk, 11.481(31) "moderate," 7.407%(20) "high," and 1.111%(3) "unclear." Most studies exhibited a low risk of selection bias (81.481%), with a few showing moderate (12.222%), high (1.852%), or unclear risk levels (4.444%) (See S6 Table).

The analysis shows that most dimensions have a substantial proportion of studies rated as low risk of bias. For example, dimensions like Research Question, Goals, and Scope (99.3% low risk), Model Structure and Assumptions (95.6% low risk), and Model Assessment (95.9% low risk) demonstrate strong methodological rigor in most studies. This suggests a robust foundation in defining research goals, constructing the model, and assessing the model's adequacy. Attrition Bias stands out with only 57.4% of studies rated as low risk, indicating that missing data could affect the reliability of findings in almost half of the studies. Sensitivity and Stability Analyses have 22.6% of studies rated as moderate risk, indicating that some studies lack careful testing of their models' robustness to varying parameters. Additionally, Selection Bias (12.2% moderate risk) and Transparency (20% moderate risk) show that there are methodological concerns in representing data selection adequately and ensuring model transparency.

The developed criteria for assessing the risk of bias is a combination of ROBINS-I [21] and principles for modeling studies suggested by [20], making it comprehensive and detailed. The detailed risk-of-bias assessment highlighted that while a substantial number of studies exhibited strong methodological rigor, a considerable proportion still faced challenges related to bias. Future studies should ensure that data sources are representative of the target population. Understanding these biases is essential for interpreting the findings accurately and for guiding future research efforts. The adoption of standardized reporting guidelines for modeling studies can improve transparency and reproducibility. Future studies should clearly distinguish between different types of interventions, such as behavioral measures and vaccination, to accurately assess their individual and combined effects.

## Limitations

The studies included in this review used different outcome measures such as COVID-19 cases, deaths, predicted number of cases, and contact tracing app usage, making direct comparison of studies challenging. Even though we collected and assessed numerical data for one of the outcome measures, the basic reproduction number ($R_0$), we only collected textual data for the other outcome measures such as COVID-19 number of cases and deaths, COVID-19 case rate, and death rates. The lack of comprehensive numerical data on these textual outcome measures in our systematic review makes it impossible to conduct a meta-analysis. This limitation restricts our ability to quantitatively synthesize the evidence and draw more precise conclusions regarding the effectiveness of the interventions.

**Table 6. Amendment/version history.**

| Version | Date | Summary of Changes | Rationale |
|---|---|---|---|
| 1.0 | December 28, 2023 | Original protocol submission. | Initial version. |
| 1.1 | September 01, 2024 | Revised the focus of the protocol to specifically target COVID-19 instead of pandemics in general. Updated the primary outcomes to include the basic reproduction number ($R_0$) as a primary outcome. Enhanced the risk of bias assessment criteria by combining principles from the guidance provided by [20] in their report entitled "Guidance for the Conduct and Reporting of Modeling and Simulation Studies in the Context of Health Technology Assessment. Methods Guide for Comparative Effectiveness Reviews. (Prepared by the Tufts Evidence-based Practice Center under Contract No. 290–2007-10055-I.)" with the ROBINS-I tool. Added more detailed data extraction items and examples for clarity. Updated the synthesis approach to include potential meta-regression for subgroups if data allows. | To increase the specificity and relevance of the review to COVID-19. Reflect on the importance of $R_0$ in assessing intervention effectiveness. Improve the rigor of risk of bias assessment and clarify the data extraction and synthesis process. |

One limitation of this study is the lack of data on specific COVID-19 variants (such as Alpha, Delta, and Omicron) associated with each reproduction number estimate. As the COVID-19 pandemic evolved, different variants emerged with varying transmissibility, which likely influenced the reproduction number $R_0$ over time. Without variant-specific data, it is challenging to account for these differences fully, and this could introduce variability into the $R_0$ estimates reported across studies. Future research that incorporates variant-specific information could provide a clearer understanding of how different strains of the virus impacted transmission dynamics.

Although we attempted to make the review process as comprehensive and inclusive as possible, there are a few aspects that may have impacted this. Most notably, our focus on publications in English and the possibly subjective interpretation of non-numerical findings may have limited the objectivity and consistency of our findings.

## Policy implications

As showcased in the studies by [32, 38], mathematical modeling plays a crucial role in informing policy decisions for COVID-19 control. It is essential that the whole modelling process, from data collection up to open-access publication, is as transparent as possible.

Models can be used to simulate the effectiveness of various control measures before real-world implementation. This allows policymakers to make informed decisions about resource allocation and intervention prioritization.

Modeling studies can highlight areas where data is limited or specific control measures require further investigation. This helps policymakers prioritize research funding and strategies to address these knowledge gaps.

Model-based insights can be translated into clear communication strategies for the public. This allows policymakers to effectively convey the benefits and importance of adhering to control measures.

## Other information

### Registration and protocol

The systematic review has been registered on the Open Science Framework (OSF) as part of the BePrepared Consortium project under the title "Modelling the impact of behavioural interventions during pandemics" (https://osf.io/q425x/).

The protocol for this systematic review has been registered on the Open Science Framework (https://osf.io/qakxz/). This registry facilitates the preregistration of research protocols to enhance transparency and credibility. The amendments made are explained in Table 6.

## Supporting information

**S1 Fig. Percentage of risk levels for reporting bias.** This figure shows the distribution of risk levels for reporting bias across the included studies. The risk levels are categorized as 'Low,' 'Moderate,' 'Unclear,' and 'High,' with corresponding percentages of 88.5%, 7.4%, 3.0%, and 1.1%, respectively. The risk levels are represented by green, yellow, gray, and red bars in the bar chart.
(TIF)

**S2 Fig. Risk of bias assessment for each study (Chunks 1, 2, 3).** This figure presents a visual summary of the risk of bias assessment for studies included in Chunks 1, 2, and 3. The left panel (Chunk 1) represents the risk of bias assessment where all 50 articles have a low risk of bias. The assessment covers 15 domains, including 'Research Question, Goals, and Scope,' 'Model Structure and Assumptions,' 'Data Informed Model,' 'Sensitivity and Stability Analyses,' and others. Judgments are categorized as 'Low,' 'Moderate,' 'High,' or 'Unclear' risk of bias, and are represented by green, yellow, red, and gray symbols, respectively.
(TIF)

**S3 Fig. Risk of bias assessment for 135 studies with all the 25 high-risk studies (Chunks 4, 5, 6).** This figure presents a visual summary of the risk of bias assessment for studies included in Chunks 4, 5, and 6. The assessment is based on various domains, including 'Research Question, Goals, and Scope,' 'Model Structure and Assumptions,' 'Data Informed Model,' 'Sensitivity and Stability Analyses,' and others. Judgments are categorized as 'Low,' 'Moderate,' 'High,' or 'Unclear' risk of bias, and are represented by green, yellow, red, and gray symbols, respectively.
(TIF)

**S4 Fig. Network visualization of the terms "preventive behavior" and "app".** This figure presents a network visualization showing the relationships between the terms "preventive behavior" and "app." The term "preventive behavior" (left) is linked with "fear," "knowledge," and "participant." The term "app" (right) is linked with terms including "close contact," "testing," "digital contact," and "contact".
(TIF)

**S5 Fig. Network visualization of the terms "SEIR model" and "SIR model".** This figure displays network visualizations illustrating the relationships between terms associated with the "SEIR model" and "SIR model." In the left panel, the "SEIR model" is linked with "parameter," "disease," "compartment," "prediction," and "contagion." In the right panel, the "SIR model" is connected to similar terms, emphasizing its relationship with "disease," "parameter," "prediction," and "treatment".
(TIF)

**S1 Appendix. Search strategy.** The search strategy consisting search keywords.
(PDF)

**S2 Appendix. PRISMA 2020 checklist.** This is the PRISMA checklist used in this systematic review.
(PDF)

**S3 Appendix. PRISMA 2020 for abstracts checklist.** This is the PRISMA checklist for abstracts used in this systematic review.
(PDF)

**S1 Table. Extraction items, descriptions, and possible values.** This table represents extraction items, their descriptions, and the possible values.
(PDF)

**S2 Table. Principles for good practice in modeling and simulation and domains from the ROBINS-I (Risk of bias In non-randomized studies—of Interventions) tool (The Netherlands, 2024).** This table represents the description of the 15 risks of bias assessment domains where eight are from the Principles for good practice in modeling and simulation and the rest seven are from the ROBINS-I.
(PDF)

**S3 Table. Descriptive statistics of categorical variables before the risk of bias assessment.**
(PDF)

**S4 Table. Summary statistics of the regression model parameters and 95% confidence intervals including sample size variable.**
(PDF)

**S5 Table. Variance Inflation Factors (VIF) for predictors in the model to check for multicollinearity.**
(PDF)

**S6 Table. Frequency table for each domain of the risk of bias (ROB) assessment.**
(PDF)

**S7 Table. An overall rating of the risk of bias assessments.**
(PDF)

**S1 Dataset. A numbered excel file of all studies identified in the literature search.** This dataset includes the titles of all identified studies and is available under the name "TKG-CJAallSearch_15781".
(XLSX)

**S2 Dataset. The 7, 616 articles screened, including those excluded from the analyses.** This dataset contains a column specifying the reasons for exclusion or inclusion and is provided in Excel format under the name "All_7616_Articles_w_Reason".
(XLSX)

**S3 Dataset. An excel file of all extracted data.** This dataset includes key information extracted from the studies and is available under the name "TKG-CJA-ExtractedData_406_270".
(XLSX)

**S4 Dataset. An excel file showing the completed risk of bias assessments for each study.** This dataset includes domain-level and overall bias assessments and is provided under the name "270ROBexcel_Nov11cor".
(XLSX)

## Acknowledgments

We thank Marijn de Bruin (Radboud UMC, Nijmegen), Luc Coffeng (Erasmus UMC Rotterdam), Sake de Vlas (Erasmus University, Rotterdam) and Peter Lugtig (Utrecht University,

Utrecht) for helpful discussions during the systematic review phase. We also thank J.D. (Joost) Driesens, an Academic Information Specialist from the University of Groningen Library, for providing feedback on the number of databases used, search strategies, and reference management tools.

## Author Contributions

**Conceptualization:** Casper Albers.

**Data curation:** Tsega Kahsay Gebretekle.

**Formal analysis:** Tsega Kahsay Gebretekle.

**Funding acquisition:** Casper Albers.

**Investigation:** Tsega Kahsay Gebretekle, Casper Albers.

**Methodology:** Tsega Kahsay Gebretekle, Casper Albers.

**Project administration:** Casper Albers.

**Resources:** Tsega Kahsay Gebretekle.

**Software:** Tsega Kahsay Gebretekle.

**Supervision:** Casper Albers.

**Validation:** Tsega Kahsay Gebretekle.

**Visualization:** Tsega Kahsay Gebretekle.

**Writing – original draft:** Tsega Kahsay Gebretekle.

**Writing – review & editing:** Tsega Kahsay Gebretekle, Casper Albers.

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
