## [Decision Letter · Decision Letter 0]

4 Oct 2024

PONE-D-24-38089Modelling the impact of behavioural interventions during pandemics: A systematic review.PLOS ONE

Dear Dr. Gebretekle,

Thank you for submitting your manuscript to PLOS ONE. After careful consideration, we feel that it has merit but does not fully meet PLOS ONE’s publication criteria as it currently stands. Therefore, we invite you to submit a revised version of the manuscript that addresses the points raised during the review process.

We look forward to receiving your revised manuscript.

Kind regards,

Martial L Ndeffo-Mbah, Ph.D

Academic Editor

PLOS ONE

3. As required by our policy on Data Availability, please ensure your manuscript or supplementary information includes the following:

Additional Editor Comments:

The manuscript require a in-depth revision from both a methodological and a results standpoint. Systematic review generally requires a critical appraisal of the included studies, but this seems to be lacking. Morover, the analysis seems to be mostly focused around the reproductive number (basic and effective), so the scope of the study may need to be refined to enphasize its focus and limitation to these metrics.

Reviewers' comments:

Reviewer's Responses to Questions

**Comments to the Author**

1. Is the manuscript technically sound, and do the data support the conclusions?

Reviewer #1: Yes

Reviewer #2: Yes

Reviewer #3: Partly

Reviewer #4: Yes

2. Has the statistical analysis been performed appropriately and rigorously? 

Reviewer #1: Yes

Reviewer #2: Yes

Reviewer #3: I Don't Know

Reviewer #4: Yes

3. Have the authors made all data underlying the findings in their manuscript fully available?

Reviewer #1: Yes

Reviewer #2: Yes

Reviewer #3: Yes

Reviewer #4: Yes

4. Is the manuscript presented in an intelligible fashion and written in standard English?

Reviewer #1: Yes

Reviewer #2: Yes

Reviewer #3: Yes

Reviewer #4: No

5. Review Comments to the Author

Reviewer #1: The authors conducted a systematic review in accordance with PRISMA 2020 guidelines to explore transmission models and assess the impact of behavioral interventions on COVID-19 outcomes. They included multiple peer-reviewed studies focusing on human subjects and modeling behavioral interventions during the pandemic, utilizing real data from various geographical regions. The methodologies employed in the studies are clearly articulated, and the presentation of the findings is commendable. Results are presented in detail and enhanced with visual representations. This thorough review underscores the significance of behavioral interventions in mitigating COVID-19 transmission and identifies opportunities for enhancing the transparency and rigor of future research.

This work can be accepted for publication. However, there is one point on which a brief discussion will be appreciated:

While a thorough descriptive and narrative analysis is described in the study, can you provide some rationale on the variation in the reproduction number across countries/ continents? From this systematic review, what is that major takeaway in this context? Why could there be this variability? Will data quality and analysis have an impact on this?

Reviewer #2: Comments to the Author

Overall, this paper is important for health behavioral science. I appreciated the effort that the authors put in to examine this research question that will undoubtably be important for future pandemics. The additional risk of bias assessment was interesting and I think adds another dimension to this manuscript that is not always addressed in the literature. I think this manuscript it written well and is important for outbreak investigation science and epidemiology.

Methods

General: The analyses and dedication to reporting as much as possible is commendable for such a large amount of data. I think the different analyses and acknowledgement and transparency of methodology is great.

Specific Line-by-Line Comments

Line 49: Capitalize “Behavioral” at the beginning of this sentence to match the other bullet points

Line 537: The term ‘eradicate’ may not be appropriate here…I think ‘eliminate’ is a better choice.

Lines 537 and 543: Starting a sentence with the reference number does not seem appropriate.

Line 574: The 0 in the R0 term is not subscripted here.

Line 693: Why is there an upside-down exclamation point here? Is this supposed to be ‘60%’?

Line 702: Missing a quotation mark after the word lockdown.

Line 745: Why are some of these terms using double quotations and some have single quotations? Was this on purpose, and if yes, what is the reason?

Tables

Table 4 is very very interesting. I think this table particularly will be very important for future analyses and pandemic research.

Figures

I don’t know if this is a function of the PDF I downloaded to review the article—but the figures are not highest resolution and somewhat blurry. Please ensure that all of the figures are the correct resolution when submitting.

Figure 11 is very difficult to read…..if there is a way to enlarge it or make it much higher quality, that is necessary.

I would also consider decreasing the number of figures—there are a couple that are also in Table form within the manuscript that repeat information. Although this is ore of a personal choice, I think the number of figures can be reduced.

Reviewer #3: This paper has an interesting premise, which would be a good addition to the literature. However, there are substantial issues that should be addressed before publication. These include major inconsistencies and contradictions throughout, results presented without methods, many methods that are insufficiently detailed for reproducibility, practices or methods that do not adhere to systematic review best practices, and a lack of clarity that appears to be due to unfamiliarity with infectious disease modeling. Furthermore, the primary stated objectives were not addressed.

Reviewer #4: This systematic review aims to synthesize available transmission modeling literature on the impact of behavioral interventions on COVID-19 outcomes. The authors present variation in the 1) basic reproduction number and 2) effective reproductive number after intervention, both among all included studies and stratified by geographic region, and describe the impact of behavioral interventions on other COVID-19 outcomes. They also characterize the landscape of included publications, including the type of model used and type of behavioral intervention explored. Finally, they assess the risk of bias of included publications across multiple criteria.

Strengths of this analysis include the authors' comprehensive literature search, multifaceted approach to synthesizing results from heterogeneous studies, and risk of bias assessment. The methods are described in detail and analyses appear to have been performed to a high technical standard.

A weakness is the regression modeling analysis of predictors of R0. The authors should revise the introduction section to be more focused and add citations for comparisons with existing literature in the discussion. In general, suggest revising the manuscript to be more concise.

MAJOR ISSUES

In the introduction, suggest zooming in on the focus of the paper more quickly.

-For example, lines 6-8 may be true, but are distracting as the paper is not about building a comprehensive social safety net.

-Lines 12-17: Meaning of social and behavioral indicators is unclear. Also the paper does not discuss how to build models or conduct modeling: suggest revising to motivate study objectives.

Regression analysis results (starting on line 500). Inclusion of other variables would likely make this analysis more informative.

-Lines 307-309, say "We considered the following independent variables: population consideration, compartmental, type of data, sample size, study design, and study location (continent)." Why didn't the final model include variables such as sample size and type of population?

-Table 5 and lines 508-516: Please include 95% confidence intervals for coefficients.

Discussion section lacks citations and does not effectively integrate results with existing literature.

-Lines 830-834: SEIR models are particularly effective compared to what? Regression models? SIR models? Network models? I agree that compartmental models are useful, but they are not necessarily the best choice in all situations.

-850-851: Need citations for previous studies.

-855: Need citations for other global studies

-859: Again, need citations for other studies.

-Lines 863-864: Need citations.

-Lines 867-873: This paragraph is repeating vs. interpreting the results.

-Lines 875-896: This paragraph repeatedly mentions "some studies". Suggest making this more quantitative.

Generally not concise, which distracts from the main message. Suggest editing for concision and clarity.

MINOR ISSUES

-Line 15: Please define "social and behavioral interventions" when first mentioned. You give examples of interventions, but not until line 49.

-Also, suggest standardizing which interventions you are considering across lines 45-46, 50-51, and 70-71.

-Lines 53-54: What does "with practice, without practice" mean in this context?

-Lines 201-202: Suggest referencing this supporting information above where the eight principles and seven domains are first mentioned.

-Lines 207-208: Unclear what you mean by "we synthesized the effect measures based on the qualitative and quantitative data reported in the studies." Suggest either specifying examples such as percent reduction here, or removing this part of the sentence and describing in later bullets.

-Table 1: Please clarify what you mean by "modeling and simulation study" vs. "predictive modeling study".

-Figures 7-8: Please list continents in the same order for these two figures. Also, might be helpful to include sample size counts in Figure 8.

-Lines 416-417: Do you have evidence/citations that the countries listed (US, China, India, Brazil, Italy) have large national science budgets? Might be true but not discussed. Also, this might be better in discussion.

-Figure 6: How did you combine the individual bias domains into an overall bias score?

-Figure 11: How did you decide which terms to highlight in this figure?

-Line 768: I suspect that sample size may also be a main contributor to the variability.

OTHER POINTS

-Initials in the financial disclosure don't match any of the authors' names.

6. PLOS authors have the option to publish the peer review history of their article (what does this mean?). If published, this will include your full peer review and any attached files.

Reviewer #1: No

Reviewer #2: No

Reviewer #3: No

Reviewer #4: No

---

## [Author Response · Author response to Decision Letter 0]

18 Nov 2024

We have provided all responses to reviewer and editor comments in the ''Response to Reviewers_CJA_TKG.pdf'' file.

---

## [Decision Letter · Decision Letter 1]

16 Dec 2024

PONE-D-24-38089R1Modelling the impact of behavioural interventions during pandemics: A systematic review.PLOS ONE

Dear Dr. Gebretekle,

Thank you for submitting your manuscript to PLOS ONE. After careful consideration, we feel that it has merit but does not fully meet PLOS ONE’s publication criteria as it currently stands. Therefore, we invite you to submit a revised version of the manuscript that addresses the points raised during the review process.

We look forward to receiving your revised manuscript.

Kind regards,

Martial L Ndeffo-Mbah, Ph.D

Academic Editor

PLOS ONE

Journal Requirements:

Additional Editor Comments :

Address the minor issues raised by reviewer #3, before we can proceed further with your manuscript.

Reviewers' comments:

Reviewer's Responses to Questions

**Comments to the Author**

1. If the authors have adequately addressed your comments raised in a previous round of review and you feel that this manuscript is now acceptable for publication, you may indicate that here to bypass the “Comments to the Author” section, enter your conflict of interest statement in the “Confidential to Editor” section, and submit your "Accept" recommendation.

Reviewer #1: All comments have been addressed

Reviewer #2: All comments have been addressed

Reviewer #4: (No Response)

2. Is the manuscript technically sound, and do the data support the conclusions?

Reviewer #1: Yes

Reviewer #2: Yes

Reviewer #4: Yes

3. Has the statistical analysis been performed appropriately and rigorously? 

Reviewer #1: Yes

Reviewer #2: Yes

Reviewer #4: Yes

4. Have the authors made all data underlying the findings in their manuscript fully available?

Reviewer #1: Yes

Reviewer #2: Yes

Reviewer #4: Yes

5. Is the manuscript presented in an intelligible fashion and written in standard English?

Reviewer #1: Yes

Reviewer #2: Yes

Reviewer #4: Yes

6. Review Comments to the Author

Reviewer #1: The work can be accepted for publication.

The authors have provided detailed explanation the the comments raised and incorporated the suggestions.

Reviewer #2: Thank you for addressing my comments. I find the revised version acceptable for publication. This paper will add a significantly important piece of the puzzle to understanding pandemic-level respiratory public health interventions.

Reviewer #4: Thank you for all your work to revise the manuscript and respond to suggestions. I appreciate your explanation for the regression model selection process. A few last minor recommendations for the "Regression analysis results for R0" section:

-When mentioning significant F-statistics, please state the null hypothesis (which model are you comparing to).

-Note on presenting AIC/BIC in your tables: these are useful for comparing between models. A single value on its own is not interpretable.

-Suggest removing text about statistical significance of intercept (lines 565-566 in revised manuscript).

7. PLOS authors have the option to publish the peer review history of their article (what does this mean?). If published, this will include your full peer review and any attached files.

Reviewer #1: No

Reviewer #2: No

Reviewer #4: No

---

## [Author Response · Author response to Decision Letter 1]

17 Dec 2024

The responses to specific reviewer comments are attached as a file in the rebuttal letter file labeled as 'Response to Reviewers_CJA_TKG2'.

---

## [Editor Report · Decision Letter 2]

20 Dec 2024

Modelling the impact of behavioural interventions during pandemics: A systematic review.

PONE-D-24-38089R2

Dear Dr. Gebretekle,

We’re pleased to inform you that your manuscript has been judged scientifically suitable for publication and will be formally accepted for publication once it meets all outstanding technical requirements.

Kind regards,

Martial L Ndeffo-Mbah, Ph.D

Academic Editor

PLOS ONE
---

## [Editor Report · Acceptance letter]

7 Jan 2025

PONE-D-24-38089R2 

PLOS ONE

Dear Dr. Gebretekle, 

I'm pleased to inform you that your manuscript has been deemed suitable for publication in PLOS ONE. Congratulations! Your manuscript is now being handed over to our production team.

Kind regards, 

on behalf of

Dr. Martial L Ndeffo-Mbah 

Academic Editor

PLOS ONE